# Lurie Networks with Robust Convergent Dynamics

**Carl R. Richardson**[1, 2]                                    *cr2g16@soton.ac.uk*

**Matthew C. Turner**[1]                                        *m.c.turner@soton.ac.uk*

**Steve R. Gunn**[1]                                            *srg@ecs.soton.ac.uk*

[1] *School of Electronics and Computer Science, University of Southampton, Southampton, UK*
[2] *The Alan Turing Institute, London, UK*

**Reviewed on OpenReview:** *https://openreview.net/forum?id=3Jm4dbrKGZ*

## Abstract

The Lurie network is a novel and unifying time-invariant neural ODE. Many existing continuous-time models, including recurrent neural networks and neural oscillators, are special cases of the Lurie network in this context. Mild constraints on the weights and biases of the Lurie network are derived to ensure a generalised concept of stability is guaranteed. This generalised stability measure is that of $k$-contraction which permits global convergence to a point, line or plane in the neural state-space. This includes global convergence to one of multiple equilibrium points or limit cycles as observed in many dynamical systems including associative and working memory. Weights and biases of the Lurie network, which satisfy the $k$-contraction constraints, are encoded through unconstrained parametrisations. The novel stability results and parametrisations provide a toolset for training over the space of $k$-contracting Lurie network's using standard optimisation algorithms. These results are also leveraged to construct and train a graph Lurie network satisfying the same convergence properties. Empirical results show the improvement in prediction accuracy, generalisation and robustness on a range of simulated dynamical systems, when the graph structure and $k$-contraction conditions are introduced. These results also compare favourably against other well known stability-constrained models and an unconstrained neural ODE.

## 1 Introduction

A Lurie[1] system is a class of nonlinear ordinary differential equations (ODE) comprising a linear time-invariant (LTI) component interconnected with a, potentially time-varying, nonlinearity. Such systems are ubiquitous throughout the sciences and engineering, including machine learning (ML) and neuroscience Pauli et al. (2021); Lessard et al. (2016); Lanthaler et al. (2024); Wilson & Cowan (1972). When modelling time-invariant dynamical systems, many ML models including linear state space models (LSSM), recurrent neural networks (RNN) and some graph neural networks (GNN) are special cases of a Lurie system (§3.1, §B.2). Such models have proven to be highly expressive as demonstrated by their successful application on a wide range of tasks such as sequential processing Kozachkov et al. (2022a); Erichson et al. (2020); Chang et al. (2019); Gu et al. (2021), computer vision Erichson et al. (2020); Chang et al. (2019); Gu et al. (2021), language modelling Gu et al. (2021) and computational chemistry Rusch et al. (2022).

Consider dynamical systems in neuroscience. The brain organises its representations of the world and carries out complex functions through collective interactions of simpler modules Kandel et al. (2000). More abstractly, this can be viewed as a graph structured dynamical system. Convergent dynamics in widespread regions of the central nervous system are thought to play a crucial role in: forming some of these representations Khona

---

[1]Named after Anatolii Isakovich Lurie and sometimes spelt Lur'e or Lurye.

& Fiete (2022), processing information over extended periods Vogt et al. (2022), learning Kozachkov et al. (2020); Centorrino et al. (2022), memory storage Hopfield (1982; 1984); Kozachkov et al. (2022b); Pals et al. (2024) and enhancing the robustness of each of these functions Khona & Fiete (2022). As summarised in Khona & Fiete (2022), convergent dynamics in the brain take several forms. For example, neural circuits with multiple equilibrium points (bistable and multi-stable) have been observed in the anterlor lateral motor cortex of a rat Piet et al. (2017) and are conjectured to appear in the mammalian hippocampus and auditory cortex. Some theories of associative memory also believe reconstruction of a learned pattern is obtained by flow to equilibrium points Krotov & Hopfield (2020); Sharma et al. (2022); Kozachkov et al. (2023). Limit cycles are another form of convergent dynamics for which there are numerous examples within the central nervous system. These include working memory Kozachkov et al. (2022b), which is thought to arise from the sustained spiking of neurons Ashwin et al. (2016), and sleep cycle generation Adamantidis et al. (2019). The graph structure and convergent dynamics are also shared with many other dynamical systems such as chemical processes Ofir et al. (2023), opinion dynamics and power systems Ofir et al. (2024). As a result, encoding such structural and dynamical properties as an inductive bias is motivated by neuroscience, for a general ML framework, and for learning models of dynamical systems.

The convergence and stability analysis of dynamical systems has been well-studied in the control theory literature. A pertinent example is the absolute stability problem where the nonlinearity of the Lurie system is unknown, but assumed to be sector-bounded or slope-restricted. The goal is to find conditions on the model parameters which ensure the trajectories of all Lurie systems, with nonlinearities in the assumed class, uphold a chosen definition of convergence. Approaches to this problem can be classified as Lyapunov analysis Khalil (2002); Park (2002; 1997), Zames-Falb multipliers Zames & Falb (1968); Turner & Drummond (2019); Carrasco et al. (2016); Drummond et al. (2024) or $k$-contraction analysis Zhang & Cui (2013); Ofir et al. (2023; 2024). Lyapunov and Zames-Falb multiplier methods are primarily designed to analyse the convergence to an equilibrium point, whereas $k$-contraction methods analyse a variety of global convergence behaviours including convergence to points, lines and planes. As many ML models are examples of Lurie systems and many activation functions are sector-bounded or slope-restricted (Drummond et al., 2022, Table 1), much of the literature on the absolute stability problem is applicable to systems involving neural networks Pauli et al. (2021); Richardson et al. (2023; 2024); Fazlyab et al. (2020).

Although designing networks with convergent dynamics is well motivated, ensuring such a property requires constraints on the network parameters, which can be detrimental if too restrictive. With this in mind, this work focuses on: (i) using $k$-contraction analysis to derive mild constraints on the weights of the Lurie network which ensure global convergence to a point, line or plane in the neural state space, for all Lurie networks with slope-restricted activation functions; (ii) establishing unconstrained parametrisations of these conditions which allows the Lurie network to be trained using gradient based optimisation algorithms whilst limiting the search space to weights which satisfy the $k$-contraction conditions; (iii) constructing a graph Lurie network (GLN) from individual Lurie networks and deriving constraints on the graph coupling matrix which ensure the $k$-contraction property is maintained. Similar unconstrained parametrisations are also derived. We empirically test the difference in prediction accuracy, generalisation and robustness of the unconstrained Lurie network, $k$-contracting Lurie network and GLN on a range of simulated dynamical systems datasets. We also compare against other well known stability-constrained architectures, which are special cases of the Lurie network, and an unconstrained neural ODE.

## 2 Preliminaries

### 2.1 Notation

For two integers $i < j$, we define $[i, j] := \{i, i+1, \ldots, j\}$. The set of non-negative real numbers is denoted by $\Re_+$. Symmetric matrices of dimension $n$ are denoted by $\mathcal{S}^n$ with the positive definite subset denoted by $\mathcal{S}^n_+$. All other positive definite subsets are denoted by a $+$ subscript. Square diagonal matrices are denoted by $\mathcal{D}^n$ and $n \times m$ diagonal matrices are symbolised by $\mathcal{D}^{nm}$. A positive definite (semi-definite) matrix $P$ is sometimes indicated by $P \succ 0$ ($P \succeq 0$). Negative definite (semi-definite) matrices are indicated analogously. The set of $n \times n$ orthogonal and skew-symmetric matrices are respectively denoted by $\mathcal{O}(n)$ and $\text{Skew}(n)$. For $W \in \Re^{n \times m}$, the ordered singular values are represented by $\sigma_1(W) \geq \cdots \geq \sigma_{\min(n,m)}(W) \geq 0$

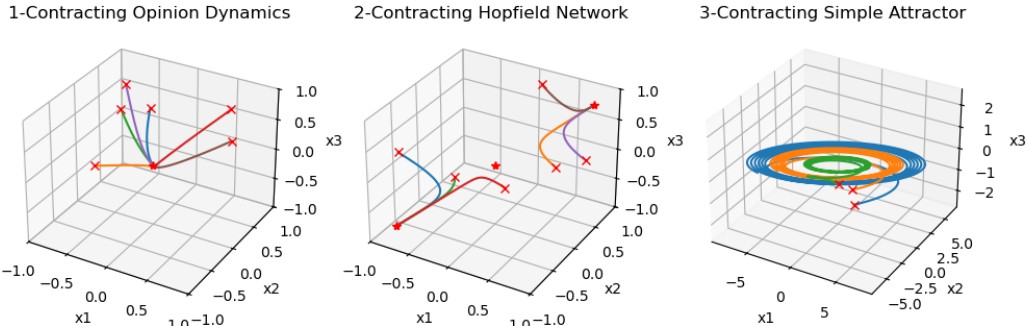

Figure 1: Trajectories from three dynamical systems satisfying the $k$-contraction property. Crosses denote the initial condition and stars denote equilibirum points.

and for $W \in \Re^{n \times n}$, the ordered eigenvalues are denoted by $\lambda_1(W) \geq \cdots \geq \lambda_n(W)$. The $k$-multiplicative and $k$-additive compound matrices of $W$ (see §A.1 for definitions) are respectively denoted by $W^{(k)}$ and $W^{[k]}$. The Jacobian of a function $f(t,x)$ is denoted by $J_f(t,x)$. The scaled 2-norm of a vector $x \in \Re^n$ with respect to (w.r.t) an invertible scaling matrix $\Theta \in \Re^{n \times n}$ is defined by $|x|_{2,\Theta} := |\Theta x|_2$, and the matrix measure induced by the scaled 2-norm is

$$\mu_{2,\Theta}(W) := \mu_2(\Theta W \Theta^{-1}) = \lambda_1\left(\frac{\Theta W \Theta^{-1} + (\Theta W \Theta^{-1})^\top}{2}\right)$$

## 2.2 $k$-contraction Analysis

In this work, we leverage $k$-contraction analysis Wu et al. (2022); Muldowney (1990), the geometrical generalisation of contraction analysis Lohmiller & Slotine (1998), as a tool for controlling convergence in the neural state space. Intuitively, $k$-contraction implies the volume of $k$-dimensional bodies exponentially converges to zero when governed by the system dynamics. Alternatively, this could be thought of as exponential convergence to a $(k-1)$-dimensional subspace. When $k = 1$, this reduces to standard contraction Lohmiller & Slotine (1998), which implies that all trajectories exponentially converge to a single trajectory. For a general time-varying dynamical system, satisfying the $k$-contraction property does not guarantee stability. However, for time-invariant dynamical systems, it has been shown that for every bounded solution: 1-contraction implies global convergence to a unique equilibrium point Lohmiller & Slotine (1998), 2-contraction implies global convergence to an equilibrium point, which is not necessarily unique but must be connected along a line Muldowney (1990), and 3-contraction, under certain assumptions, implies convergence to a non-unique attractor in a 2d subspace Cecilia et al. (2023). Three examples of $k$-contracting dynamics are presented in Figure 1.

Time-invariant dynamical systems which satisfy the $k$-contraction property for $k \in \{1, 2, 3\}$ have several desirable properties for ML models. They can exhibit a wide range of complex convergent behaviours such as multi-stable and orbitally stable systems Zoboli et al. (2024). This suggests that a model can be *expressive* whilst satisfying the $k$-contraction conditions, particularly for higher values of $k$ where the constraints are less restrictive, as highlighted in §3.2. The $k$-contraction property also implies an inherent *robustness* as the trajectories can only converge to a finite number of long term behaviours. Next, we present the fundamental $k$-contraction result from Wu et al. (2022), see §A.3 for the formal definition of $k$-contraction.

**Theorem 1** *Fix $k \in [1, n]$ and consider the nonlinear system $\dot{x} = f(t,x)$ with $f : \Re_+ \times \Re^n \to \Re^n$ continuously differentiable. If there exists $\eta > 0$ and an invertible matrix $\Theta \in \Re^{n \times n}$ such that*

$$\mu_{2,\Theta^{(k)}}\left(J_f^{[k]}(t,x)\right) \leq -\eta \quad \forall\, x \in \Re^n \text{ and } t \in \Re_+ \tag{1}$$

*then the nonlinear system is $k$-contracting in the 2-norm w.r.t the metric $P := \Theta^\top \Theta$.*

This result has two features: (i) it requires the existence of an invertible matrix $P$. In the simplest case, one can expect a solution $P = pI_n$ to exist. For other systems, such simple solutions will not exist and more general matrices such as $P \in \mathcal{S}_+^n$ will be required, making the proofs more difficult; (ii) it requires the use of compound matrices (§A.1). For a matrix $W \in \Re^{n \times m}$, the matrix $W^{[k]}$ with $k \in [1, \min(n, m)]$ will have the size $\binom{n}{k} \times \binom{m}{k}$ which is typically much larger and more computationally difficult to work with. A technical introduction to compound matrices, $k$-contraction analysis and how they relate is presented in §A. In §3.2 we derive results which verify (1) for the special case of nonlinear systems of the form (2).

## 3 Lurie Network

A *Lurie network* is defined by (2) with weights $A \in \Re^{n \times n}, B \in \Re^{n \times m}$, $C \in \Re^{m \times n}$ and biases $b_x \in \Re^n$, $b_y \in \Re^m$.

$$\dot{x}(t) = Ax(t) + B\Phi\big(y(t)\big) + b_x \qquad\qquad y(t) = Cx(t) + b_y \qquad\qquad x(0) = x_0 \qquad (2)$$

The model has a biased linear component interconnected with a nonlinearity of the form $\Phi(y) := \big[\phi_1(y_1) \ldots \phi_m(y_m)\big]^\top$ where $\phi_i(y_i)$ is assumed to be slope-restricted with an upper bound $g > 0$, such that $0 \preceq J_\Phi(y) \preceq gI_m$. This separation of the linear and nonlinear components is useful for analysis. Activation functions which satisfy this slope-restricted assumption include the hyperbolic tangent (tanh) and the rectified linear unit ($ReLU$). For simplicity, we assume the same scalar nonlinearity is applied element-wise and drop the subscript. The proposed Lurie network is a very general model as highlighted by the special cases in §3.1 and its relationship to deep feedforward neural networks as outlined in §B.1. Finally, it is important to observe that the model is time-invariant, this implies that if Theorem 1 is satisfied, the model will inherit the appealing convergence and robustness properties stated in §2.2. This may be viewed as a limitation for modelling neural dynamics since the brain is subject to various external inputs; however, it is common to assume that, at least on the time scale of interest, the dynamics evolve in a time-invariant manner Khona & Fiete (2022).

### 3.1 Example Lurie Networks

When applied to time-invariant dynamical systems, many models from the ML literature become special cases of (2). In this setting, the time-varying external inputs are replaced with trainable biases. A subset of examples are presented next with further examples included in §B.2. As the results in §3.2 apply to any model of the form (2), they can also be applied to these special cases.

**Lipschitz RNN:** A stability constrained RNN Erichson et al. (2020) where the parameters $A, C$ are expressed as a weighted sum of symmetric and skew-symmetric terms in order to control the eigenvalues of the Jacobian. The remaining components are $B = I_n$, $b_x = 0$ and $\phi(\cdot) \equiv \tanh(\cdot)$.

$$A, C \in \{(1 - \beta)(W + W^\top) + \beta(W - W^\top) - \gamma I_n \mid W \in \Re^{n \times n}, \ 0.5 \le \beta \le 1, \ \gamma > 0\}$$

**Antisymmetric RNN:** A constrained RNN Chang et al. (2019) with the same motivations as above. Related to (2) by $A = 0$, $B = I_n$, $C \in \{W - \gamma I_n \mid W \in \text{Skew}(n), \ \gamma > 0\}$, $b_x = 0$ and $\phi(\cdot) \equiv \tanh(\cdot)$.

**SVD Combo:** A 1-contracting graph coupled RNN Kozachkov et al. (2022a) where $q$ denotes the number of individual RNNs and $n$ denotes the state dimension of each RNN. A special case of (2) with $A = L - aI_{qn}$, $C = I_{qn}$ and $b_y = 0$. The graph coupling matrix is denoted by $L$ and $B$ is block diagonal where each block contains the synaptic weights of the individual RNNs. Both of these matrices are expressed by special parametrised forms to ensure the individual RNNs and graph coupled RNN are 1-contracting. As in (2), the nonlinearity is required to be slope-restricted.

### 3.2 $k$-contraction Analysis of Lurie Networks

Two sufficient results which satisfy Theorem 1 and guarantee (2) is $k$-contracting are presented next. Conditions were derived in (Ofir et al., 2024, Theorem 2) which verify Theorem 1 for a Lurie network with $A \in \mathcal{D}^n$ and $b_y = 0$. Theorem 2 extends them to account for $A \in \Re^{n \times n}$ and $b_y \neq 0$. Refer to §C.1 for the proof.

**Theorem 2** *Consider the Lurie network (2) with $\Phi(y) := \begin{bmatrix} \phi_1(y_1) & \ldots & \phi_m(y_m) \end{bmatrix}^\top$ being slope-restricted such that $0 \preceq J_\Phi(y) \preceq gI_m$. Fix $k \in [1, n]$ and define $\alpha_k := (2k)^{-1} \sum_{i=1}^{k} \lambda_i(A + A^\top)$. If $\alpha_k < 0$ and*

$$g^2 \sum_{i=1}^{k} \sigma_i^2(B)\sigma_i^2(C) < \alpha_k^2 k \tag{3}$$

*then (2) is $k$-contracting in the 2-norm w.r.t the metric $P = -\alpha_k^{-1} I_n$.*

The additional freedom permitted by $k$-contraction over standard contraction is highlighted by the summation of the eigenvalues and singular values. In 1-contraction, Theorem 2 requires the largest eigenvalue of the symmetric component of $A$ to be negative whereas for $k \in [2, n]$, this condition on $A$ becomes incrementally more relaxed as $k$ is increased. Equation (3) illustrates a similar relaxation of the constraints on $B$ and $C$. Theorem 2 has several appealing features: (i) it does not require the computation of the troublesome compound matrices; (ii) it provides a way of embedding the $k$-contraction property into the structure of a Lurie network based on fairly simple unconstrained parametrisations of the weights, as shown in §4; (iii) the biases are not present in the condition, so are naturally unconstrained; (iv) the result does not rely on symmetries of the parameters. In many Hopfield-based models of associative memory, symmetry in the parameters is needed to make the existence of a global energy function mathematically tractable; however, this simplification is biologically unrealistic and limits the model's expressive power. The limitation of the result is that only Lurie networks which are $k$-contracting in a scalar metric can be verified.

We now present a second result which addresses the scalar metric drawback; however, it comes at the cost of strong constraints on the weights $B$ and $C$. See §C.2 for the proof.

**Theorem 3** *Consider the Lurie network (2) with $\Phi(y) := \begin{bmatrix} \phi_1(y_1) & \ldots & \phi_n(y_n) \end{bmatrix}^\top$ being slope-restricted such that $0 \preceq J_\Phi(y) \preceq gI_n$. Fix $k \in [1, n]$. If $B \in \mathcal{D}^n$, $C = B^{-1}$ and*

$$P^{(k)}A^{[k]} + (A^{[k]})^\top P^{(k)} + 2kgP^{(k)} \prec 0 \tag{4}$$

*then (2) is $k$-contracting in the 2-norm w.r.t the metric $P \in \mathcal{D}_+^n$.*

Theorem 3 improves upon Theorem 2 in the sense that Lurie networks which are $k$-contracting in a diagonal metric can now be verified; however, only when $C = B^{-1}$. Due to this constraint, Theorem 3 has very limited practical use; however, it may prove to be theoretically insightful for addressing the scalar metric drawback. Finally, it is important to highlight that Theorem 2 and Theorem 3 apply to the class of slope-restricted nonlinearities, so these results address the absolute stability problem for the $k$-contraction property.

### 3.3 Graph Lurie Networks

Many larger scale dynamical systems such as molecular, social, biological, and financial networks Hamilton et al. (2017) naturally have a graph structure. To make the Lurie network more applicable to these problems, a graph coupling term is introduced to model a set of $q$ interacting Lurie networks. To illustrate this, $q$ independent Lurie networks can be modelled by

$$\dot{x}(t) = A_G x(t) + B_G \Phi\big(y(t)\big) + b_x \qquad y(t) = C_G x(t) + b_y \qquad x(0) = x_0 \tag{5}$$

where the weights are $A_G := \text{blockdiag}(A_1, \ldots, A_q) \in \Re^{qn \times qn}$, $B_G := \text{blockdiag}(B_1, \ldots, B_q) \in \Re^{qn \times qm}$, $C_G := \text{blockdiag}(C_1, \ldots, C_q) \in \Re^{qm \times qn}$ and the biases are $b_x \in \Re^{qn}$, $b_y \in \Re^{qm}$. The graph Lurie network (GLN) is then defined by

$$\dot{x}(t) = (A_G + L)x(t) + B_G \Phi\big(y(t)\big) + b_x \qquad y(t) = C_G x(t) + b_y \qquad x(0) = x_0 \tag{6}$$

where $L := \begin{bmatrix} L_{jl} \end{bmatrix}$ is a block matrix with block $L_{jl} \in \Re^{n \times n}$ connecting Lurie network $l$ to Lurie network $j$. The state and nonlinearity of the GLN are defined by $x \in \Re^{qn}$ and $\Phi : \Re^{qm} \to \Re^{qm}$ where the states of the independent Lurie networks have been stacked into a single state. Interestingly, the GLN (6) is actually a

special case of a Lurie network; however, as the networks get larger, so does the search space. Thus, imposing any assumptions which respect the structure of the problem can reduce the search space and lead to more robust and generalisable models. Any further prior knowledge about the graph can be encoded through constraints on the graph coupling matrix.

### 3.4 $k$-contraction Analysis of Graph Lurie Networks

In this section, we assume that $q$ independent Lurie networks (2) are $k$-contracting in the 2-norm w.r.t metrics $P_1, \ldots, P_q$. This is equivalent to (5) $k$-contracting in the 2-norm w.r.t the metric $P = \mathrm{blockdiag}(P_1, \ldots, P_q)$. Theorem 2 and Theorem 3 provide two results which can, respectively, verify this w.r.t a scalar metric $P_j = p_j I_n$ and a diagonal metric $P_j \in \mathcal{D}_+^n$ for $j \in [1, q]$. Other results may be used providing they apply to systems of the form (2). Under this assumption, Theorem 4 provides a constraint on the graph coupling term which ensures the GLN is $k$-contracting when constructed from $q$ independently $k$-contracting Lurie networks. The proof is detailed in §C.3.

**Theorem 4** *Fix $k \in [1, n]$. Consider a GLN (6) where the $q$ independent Lurie networks are collectively defined by the weights $A_G := \mathrm{blockdiag}(A_1, \ldots, A_q)$, $B_G := \mathrm{blockdiag}(B_1, \ldots, B_q)$, $C_G := \mathrm{blockdiag}(C_1, \ldots, C_q)$ and biases $b_x \in \Re^{qn}$, $b_y \in \Re^{qm}$ and are $k$-contracting in the 2-norm w.r.t the metric $P := \mathrm{blockdiag}(P_1, \ldots, P_q)$. If the graph coupling matrix $L \in \Re^{qn \times qn}$ satisfies*

$$P^{(k)} L^{[k]} + (L^{[k]})^\top P^{(k)} \preceq 0 \tag{7}$$

*then (6) is also $k$-contracting in the 2-norm w.r.t the metric $P$.*

**Remark 1** *It should be noted that Theorem 4 could be applied more generally for constructing $k$-contracting graph coupled systems. Providing the $q$ subsystems are $k$-contracting in the 2-norm w.r.t some metric $P = \mathrm{blockdiag}(P_1, \ldots, P_q)$, then Theorem 4 is applicable when coupling them through the graph coupling term. It is not necessary for the $q$ subsystems to be Lurie networks.*

## 4 Parametrisation of $k$-contracting Lurie Networks

To train a $k$-contracting Lurie network using gradient based optimisers, parametrisations which express the constrained weights in terms of unconstrained variables must be found. To formalise this idea, we define the sets $\Omega_2(g, k)$ and $\Omega_4(k, P)$. As the biases do not appear in these sets, they are naturally unconstrained. The set $\Omega_2(g, k)$ contains all the weights of the Lurie network which satisfy Theorem 2.

$$\Omega_2(g, k) := \left\{ (\bar{A}, \bar{B}, \bar{C}) \,\Big|\, \alpha_k := \frac{1}{2k} \sum_{i=1}^{k} \lambda_i(A + A^\top) < 0, \; z_k := g^2 \sum_{i=1}^{k} \sigma_i^2(\bar{B}) \sigma_i^2(\bar{C}) < \alpha_k^2 k \right\} \tag{8a}$$

$$\Omega_4(k, P) := \left\{ \bar{L} \in \Re^{qn \times qn} \,\Big|\, P^{(k)} \bar{L}^{[k]} + (\bar{L}^{[k]})^\top P^{(k)} \preceq 0 \right\} \tag{8b}$$

The set $\Omega_4(k, P)$ contains the graph coupling matrices which satisfy Theorem 4. A parametrisation associated with Theorem 3 was not established due to its limitations mentioned earlier. The next results present two different parametrisations of the set $\Omega_2(g, k)$. See §C.4 for the proofs which leverage the eigenvalue and singular value decompositions.

**Theorem 5** *Given $g > 0$, $k \in [1, n]$, $U_A, U_B, V_C \in \mathcal{O}(n)$, $V_B, U_C \in \mathcal{O}(m)$, $\Sigma_B \in \mathcal{D}_+^{nm}$, $\Sigma_C \in \mathcal{D}_+^{mn}$, $Y_A \in \mathrm{Skew}(n)$, $G_A \in \mathcal{D}_+^n$ and define*

$$A := \frac{1}{2} U_A \Sigma_A U_A^\top + \frac{1}{2} Y_A \qquad\qquad \Sigma_A := -\sqrt{\frac{4 z_k}{k}} I_n - G_A \tag{9a}$$

$$B := U_B \Sigma_B V_B^\top \qquad\qquad C := U_C \Sigma_C V_C^\top \tag{9b}$$

*then $(A, B, C) \in \Omega_2(g, k)$.*

**Theorem 6** *Given $g > 0$, $k \in [1, n]$, $U_A, U_B, V_C \in \mathcal{O}(n)$, $V_B, U_C \in \mathcal{O}(m)$, $\Sigma_B \in \mathcal{D}_+^{nm}$, $\Sigma_C \in \mathcal{D}_+^{mn}$, $Y_A \in \mathrm{Skew}(n)$, $\Sigma_{A1} \in \mathcal{D}^{k-1}$, $G_{A2} > 0$, $G_{A3} \in \mathcal{D}_+^{n-k}$ and define*

$$A := \frac{1}{2} U_A \Sigma_A U_A^\top + \frac{1}{2} Y_A \qquad\qquad B := U_B \Sigma_B V_B^\top \qquad\qquad C := U_C \Sigma_C V_C^\top \qquad (10a)$$

$$\Sigma_A := \mathrm{blockdiag}(\Sigma_{A1}, \Sigma_{A2}, \Sigma_{A3}) \qquad\qquad \Sigma_{A1} \in \mathcal{D}^{k-1} \qquad\qquad\qquad (10b)$$

$$\Sigma_{A2} := -\sqrt{4kz_k} - \sum_i^{k-1} (\Sigma_{A1})_{ii} - G_{A2} \qquad\qquad \Sigma_{A3} := \min(\Sigma_{A1}, \Sigma_{A2}) I_{n-k} - G_{A3} \qquad (10c)$$

*then $(A, B, C) \in \Omega_2(g, k)$.*

**Remark 2** *In both results, a mapping between the sets* Skew($\cdot$) *and* $\mathcal{O}(\cdot)$ *was exploited to express the orthogonal matrices in terms of unconstrained skew-symmetric matrices Lezcano-Casado & Martinez-Rubio (2019). The remaining variables are unconstrained or simply require positive elements, which can be obtained by taking the absolute value of unconstrained variables.*

In Theorem 5 and Theorem 6, the $B$ and $C$ matrices are unconstrained since they are simply expressed by their singular value decomposition. The variables representing the singular values of $B$ and $C$ directly upper bound $\alpha_k$. The only source of conservatism in the parametrisations is introduced through the definition of $\Sigma_A$. In Theorem 5, the constraint on $\alpha_k$ is enforced by a uniform negative constraint on the diagonal elements of $\Sigma_A$. If this assumption is true, this significantly speeds up the learning process; however, it can be prohibitive if not. Theorem 6 allows the largest $(k-1)$ eigenvalues to be unconstrained ($\Sigma_{A1}$), meaning non-Hurwitz $A$ matrices are encapsulated in the parametrisation. The constraint on $\alpha_k$ is implemented by the $\Sigma_{A2}$ block and the $\Sigma_{A3}$ block ensures the remaining eigenvalues are less than the other $k$. As $k$ is a hyperparameter of the model, it is useful to note that the set $\Omega_2(g, k-1)$ intersects with $\Omega_2(g, k)$ when the following holds

$$g^2 \sigma_k^2(B) \sigma_k^2(C) < \frac{1}{4k} \lambda_k^2(A + A^\top) + \frac{k-1}{k} \alpha_{k-1} \lambda_k(A + A^\top) - \frac{k-1}{k} \alpha_{k-1}^2 \qquad (11)$$

We expect this to hold for many realistic systems since (11) just requires $\lambda_k(A + A^\top)$ to be sufficiently negative to counteract the product of the growth of $B$ and $C$ in the $k^{th}$ direction. Hence, if the best value of $k$ is unknown for a given application, then setting $k = 3$ allows the model to search over many of the stable $l$-contracting systems parametrised by Theorem 5 or Theorem 6, where $l \in [1, k]$. See §C.5 for the proof.

The next result provides an unconstrained parametrisation of the set $\Omega_4(k, P)$ in (8b). Coupling this with the earlier parametrisations allows the $k$-contracting GLN to be trained with gradient based optimisers. See §C.6 for the proof.

**Theorem 7** *Given $k \in [1, n]$, $G_{L1}, G_{L2} \in \Re^{qn \times qn}$, $\Theta = \mathrm{blockdiag}(\Theta_1, \ldots, \Theta_q)$ where $\Theta_j \in \mathcal{S}_+^n$ for $j \in [1, q]$, $P = \Theta^\top \Theta$ and define*

$$L := G_{L1} - P^{-1} G_{L1}^\top P + \Theta^{-1} G_{L2} \Theta \qquad (12)$$

*where $(G_{L2} + G_{L2}^\top)^{[k]} \preceq 0$, then $L \in \Omega_4(k, P)$.*

Theorem 7 provides flexibility in the choice of $G_{L1}, G_{L2}$ to define different graph structures, where the constraint on $G_{L2}$ is equally a constraint on the sum of it's $k$-largest eigenvalues (Fact 7). This constraint could be parametrised in a simlar manner to the symmetric component of $A$ in Theorem 5 and Theorem 6; in which case, Theorem 7 is satisfied and the graph has a directed all-to-all structure, including self-loops.

**Remark 3** *If $G_{L2} = 0$ along with the main diagonal blocks of $G_{L1}$, then (12) is a boundary condition of (7). Thus, Theorem 7 is satisfied and the self-loops are removed from the graph structure.*

One consideration when constructing $k$-contracting Lurie networks is the number of additional parameters required for the parametrisation. A Lurie network, as defined in (2), has a total parameter count $N_L =$

$n^2 + 2nm + n + m$; whereas a $k$-contracting Lurie network, constructed according to Theorem 5 or Theorem 6, has a total parameter count $N_K \leq 2n^2 + m^2$. The number of parameters only increases significantly when $m$ is large compared to $n$; however, throughout the literature, many special cases of a Lurie network set $n = m$ (see §3.1); in which case, $N_K$ becomes marginally smaller than $N_L$. Furthermore, the all-to-all graph coupling term has $N_C = 2(qn)^2$ parameters or $N_C = qn^2(q-1)$ if the self-loops are removed according to Remark 3. This analysis highlights that ensuring the Lurie network and GLN are $k$-contracting comes at a minimal computational expense.

## 5  Empirical Evaluation

In §1, it was highlighted that a range of convergent dynamics are prevalent in many dynamical systems and neural processes. In the following section we test the impact of the additional expressivity of the Lurie network and the importance of encoding this general convergence property for modelling a range of dynamical systems. The prediction accuracy, generalisation and robustness of the proposed models were used to compare performance. Refer to Richardson et al. (2025) for additional experiments regarding the Lurie network's ability to form representations.

### 5.1  Datasets and Training

We first consider three time-invariant dynamical systems: (i) an opinion dynamics model of a social network where all opinions agree and thus converge to a unique equilibrium point; (ii) a Hopfield network of associative memory with two stable equilibrium points and one unstable; (iii) a generic simple attractor which could be used to model the stored patterns in working memory. Each system has $n = 3$ states allowing for visualisation of the ground truth and predictions. For each dynamical system, we have a dataset including $1,000$ trajectories, sampled every $0.01s$ over a $20s$ interval. The test sets were formed by holding out 100 trajectories. The input to each model was the initial condition sampled from a uniform distribution with the domain $(-1, +1)^3$ for the opinion/Hopfield datasets and $(-3, +3)^3$ for the simple attractor. The full trajectory was then used as the target to train the model. An illustration of these datasets can be seen in Figure 1 and further details about the data generation can be found in Appendix D.

To test the out of distribution generalisation and robustness, two additional datasets were generated for each of the opinion, Hopfield and attractor tasks. These differ from the training datasets in the following ways: (i) they include 100 trajectories over a $30s$ interval; (ii) the initial conditions were sampled from a uniform distribution over the intervals $1 < |x_i(0)| < 4$ for the opinion/Hopfield datasets and $3 < |x_i(0)| < 6$ for the simple attractor, where $i \in \{1, 2, 3\}$. These trajectories are $10s$ longer than the training data with initial conditions also sampled outside the training distribution. To test the robustness, noise sampled from the standard normal distribution was added to the initial conditions of one of these datasets, before generating the trajectories.

For the second set of experiments, we consider two 30 dimensional dynamical systems. The first is a graph-coupled (GC) Hopfield network, formed by connecting 10 previously described Hopfield networks through a graph coupling matrix. To ensure the convergence property was preserved, the matrix was expressed by (12) with $G_{L1}$ sampled from a uniform distribution and $G_{L2} = 0$. The second system was a graph-coupled attractor, constructed in the same way. The datasets were generated as described in the previous two paragraphs; however, they each included $30,000$ trajectories.

The training settings used are explicitly detailed in Appendix D. For all models and all datasets, the mean squared error (MSE) loss was used alongside the Adam optimiser. All code was implemented in PyTorch and can be found at https://github.com/CR-Richardson/LurieNetwork.

### 5.2  $k$-contracting Lurie Network

In this section, we compare the $k$-contracting Lurie network against five other continuous-time models: (i) the unconstrained Lurie network, for testing the importance of the $k$-contraction constraints; (ii) an unconstrained neural ODE with two hidden layers, each comprised of 20 neurons and *ReLU* activations; (iii) the three

constrained continuous-time models detailed in §3.1 where the SVD combo network is comprised of a single node. The numerical integration was performed using the Euler method with step size $\delta = 1 \times 10^{-2}$. Besides the neural ODE, each model had tanh activations, which is slope-restricted with $g = 1$.

For the opinion and Hopfield datasets, the $k$-contracting Lurie network was constructed according to Theorem 5 whereas Theorem 6 was used for the simple attractor. We set $k = 1$ for the opinion dynamics, $k = 2$ for the mulit-stable Hopfield network and $k = 3$ for the simple attractor. Similar results were obtained when setting $k = 3$ for all examples.

Table 1: MSE on a test set of 100 trajectories. The lowest MSE is presented alongside the mean and standard deviation calculated after training each model $N = 3$ times on a single T4 GPU (Google Colab).

| Model | MSE (mean $\pm$ std, best) | | |
|---|---|---|---|
| | Opinion | Hopfield | Attractor |
| $k$-Lurie Network | $(8.0 \pm 3.0, 5.10) \times 10^{-5}$ | $(1.5 \pm 1.0, \mathbf{0.26}) \times 10^{-2}$ | $(3.5 \pm 1.0, \mathbf{1.70}) \times 10^{-3}$ |
| Lurie Network | $(3.7 \pm 4.0, 0.53) \times 10^{-3}$ | $(3.6 \pm 2.0, 0.39) \times 10^{-1}$ | $(5.1 \pm 5.0, 0.57) \times 10^{-1}$ |
| Neural ODE | $(2.0 \pm 2.0, \mathbf{0.43}) \times 10^{-4}$ | $(2.5 \pm 1.0, 1.50) \times 10^{-2}$ | $(2.0 \pm 1.0, 1.00) \times 10^{-2}$ |
| Lipschitz RNN | $(3.9 \pm 3.0, 0.88) \times 10^{-2}$ | $(2.9 \pm 2.0, 0.30) \times 10^{-1}$ | $1.48 \pm 2.0, 1.10 \times 10^{-2}$ |
| SVD Combo | $(8.6 \pm 3.0, 5.70) \times 10^{-4}$ | $(2.9 \pm 0.6, 2.10) \times 10^{-1}$ | $3.12 \pm 0.5, 2.74$ |
| Antisym. RNN | $(30.0 \pm 0.2, 28.0) \times 10^{-2}$ | $(4.3 \pm 0.2, 4.11) \times 10^{-1}$ | $6.93 \pm 0.2, 6.74$ |

Table 1 compares the MSE on the test set of each task. The $k$-contracting Lurie network achieved the best MSE on two out of three examples. The importance of the $k$-contraction conditions is particularly clear when comparing the mean and standard deviation with that of the unconstrained Lurie network. These conditions clearly reduce the search space to a tractable region to optimise over as the MSE of the unconstrained Lurie network is at least an order of magnitude worse than its $k$-contracting counterpart. The other models perform as one would expect: (i) the neural ODE demonstrates strong accuracy across all tasks; (ii) SVD combo performs well on the opinion dataset, where the 1-contraction assumption is valid, but struggles on the others; (iii) the Lipschitz RNN struggles on the attractor dataset whilst the antisymmetric RNN struggles across the board due to the A matrix being fixed at zero and the eigenvalues of the C matrix being fixed to almost purely imaginary values. Figures 3, 6, 9 show a random sample of trajectories from each test set, along with the predictions of each model.

Table 2: MSE on a new test set of 100 trajectories where: (i) the initial conditions were sampled outside the range used for training and the trajectories were $10s$ longer than the training set; (ii) additionally, noise sampled from the standard normal distribution was added to the initial conditions.

| Model | MSE | | | MSE (noisy inputs) | | |
|---|---|---|---|---|---|---|
| | Opinion | Hopfield | Attractor | Opinion | Hopfield | Attractor |
| $k$-Lurie Network | $\mathbf{2.9 \times 10^{-3}}$ | $\mathbf{5.6 \times 10^{-2}}$ | $\mathbf{2.3 \times 10^{-1}}$ | $\mathbf{2.0 \times 10^{-2}}$ | $\mathbf{3.2 \times 10^{-1}}$ | $\mathbf{1.28}$ |
| Lurie Network | $6.4 \times 10^{-2}$ | $1.8 \times 10^{-1}$ | $5.96$ | $2.7 \times 10^{-1}$ | $4.4 \times 10^{-1}$ | $6.79$ |
| Neural ODE | $1.2 \times 10^{-2}$ | $1.09$ | $2.31$ | $3.9 \times 10^{-2}$ | $1.63$ | $4.76$ |
| Lipschitz RNN | $2.3 \times 10^{-1}$ | $7.3 \times 10^{-1}$ | $6.2 \times 10^{-1}$ | $2.9 \times 10^{-1}$ | $9.7 \times 10^{-1}$ | $1.71$ |
| SVD Combo | $1.0 \times 10^{-2}$ | $2.38$ | $20.9$ | $3.3 \times 10^{-2}$ | $7.97$ | $30.30$ |
| Antisym. RNN | $6.43$ | $5.25$ | $52.1$ | $7.29$ | $6.33$ | $52.9$ |

Table 2 compares the generalisation and robustness of the models on each task. No new models were trained, instead the best models from Table 1 were directly applied to these out of distribution and noisy datasets (§5.1). The $k$-contracting Lurie network performs the best on all of these datasets and for some, it still demonstrates a MSE of an order of magnitude lower than the next best model. The MSE of the unconstrained Lurie network and neural ODE tended to drop off for these datasets whereas the MSE of the constrained models, excluding the antisymmetric RNN, tended to stay fairly consistent when their assumptions were valid. Figures 4, 7, 10 show a random sample of noise-free trajectories and predictions for each dataset, whilst Figures 5, 8, 11 repeat the same for the noisy inputs. The $k$-contracting Lurie network predicted the

correct long-term behaviour most accurately under all conditions; even when noise was added, the error was predominantly present during the initial transient.

### 5.3 $k$-contracting Graph Lurie Network

This section repeats the same experiments as the previous section, but for two 30-dimensional graph-coupled (GC) dynamical systems: the GC Hopfield network and the GC simple attractor (§5.1). The state of these datasets is significantly larger than those used in other dynamical systems datasets such as: (i) the LASA dataset Lemme et al. (2015) where the 2-d trajectories are typically stacked to form 4-d or 8-d trajectories; (ii) simulated datasets of the 2, 4 or 8 link pendulums which, respectively, have 4, 8 or 16 dimension trajectories.

For both datasets, the GLN was constructed according to Theorem 7 and Remark 3 where $n = m = 3$ and $q = 10$. The individual Lurie networks were constructed according to Theorem 5 for the GC Hopfield network, with $k = 2$ and Theorem 6 for the GC attractor, with $k = 3$. The neural ODE was formed using two layers with 100 neurons and ReLU activations. The SVD combo leveraged a similar graph structure to the one used in this paper. The other models were the same as in the previous section but with a $(qn)$-dimensional state.

Table 3: MSE on a test set of 100 trajectories. The lowest MSE is presented alongside the mean and standard deviation calculated after training each model $N = 3$ times on a single A100 GPU (Google Colab).

| Model | MSE (mean $\pm$ std, best) | |
|---|---|---|
| | GC Hopfield | GC Attractor |
| GLN | $0.016 \pm 0.0003$, **0.016** | $0.293 \pm 0.1969$, **0.015** |
| $k$-Lurie Network | $0.238 \pm 0.0011$, 0.237 | $0.737 \pm 0.0108$, 0.723 |
| Lurie Network | $2.537 \pm 1.3327$, 1.157 | $291.8 \pm 191.84$, 21.83 |
| Neural ODE | $0.138 \pm 0.0291$, 0.114 | $3.445 \pm 0.3626$, 2.942 |
| Lipschitz RNN | $0.124 \pm 0.0161$, 0.105 | $0.658 \pm 0.1604$, 0.433 |
| SVD Combo | $0.339 \pm 0.1363$, 0.229 | $3.024 \pm 1.1240$, 1.435 |
| Antisym. RNN | $0.448 \pm 0.0023$, 0.444 | $6.386 \pm 0.0066$, 6.380 |

Table 3 shows the GLN had a lower MSE than all other models by a factor of 10. Comparing the GLN, $k$-contracting Lurie network and the unconstrained Lurie network highlights the improvements due to the $k$-contraction conditions and the graph structure. Table 4 also indicates that the GLN generalised and remained robust to noise, even in these high-dimensional systems. The same can be said for the $k$-contracting Lurie network which achieved the second lowest MSE on the generalisation and robustness tests. The inherent graph structure of the SVD combo may be the reason behind its improved ranking in the GC Hopfield tasks whereas the neural ODE particularly struggled to generalise for the GC attractor. Possible explanations behind the poor performance of the constrained benchmark models are suggested in the next section.

Table 4: MSE on a new test set of 100 trajectories where: (i) the initial conditions were sampled outside the range used for training and the trajectories were $10s$ longer than the training set; (ii) additionally, noise sampled from the standard normal distribution was added to the initial conditions.

| Model | MSE | | MSE (noisy inputs) | |
|---|---|---|---|---|
| | GC Hopfield | GC Attractor | GC Hopfield | GC Attractor |
| GLN | **0.08** | **1.67** | **0.26** | **2.85** |
| $k$-Lurie Network | 0.77 | 6.10 | 1.05 | 6.90 |
| Lurie Network | 20350 | 5638 | 25481 | 6368 |
| Neural ODE | 2.12 | 24.93 | 2.76 | 25.11 |
| Lipschitz RNN | 3.58 | 6.17 | 4.25 | 7.84 |
| SVD Combo | 0.84 | 11.40 | 1.07 | 12.30 |
| Antisym. RNN | 5.32 | 50.68 | 6.21 | 52.10 |

## 6 Related Work

Several constrained continuous-time RNN models exist in the ML literature. The model structure of three notable examples were presented in §3.1 as they happen to be special cases of a Lurie network, when modelling time-invariant dynamical systems. The antisymmetric RNN Chang et al. (2019) and the Lipschitz RNN Erichson et al. (2020) were designed to address the exploding and vanishing gradient problem Pascanu et al. (2013). The antisymmetric RNN did so by parametrising the RNN such that the real eigenvalues of the Jacobian were zero. It achieved this by setting $A = 0$ and restricting $C$ to being skew-symmetric. Whilst this does prevent the gradients from exploding and vanishing, it restricts the dynamics which the model can learn to purely oscillatory behaviour. The Lipschitz RNN has a more relaxed parametrisation. This model constructs the $A$ and $C$ matrices such that they are both convex combinations of symmetric and skew-symmetric matrices. However, the weight of the symmetric matrix can only vary between 0 and 0.5, whereas the weight of the skew-symmetric matrix can vary between 0.5 and 1. Again, this addresses the vanishing and exploding gradient problem, but the model cannot encode dynamics which are predominately decaying or growing. Finally, the RNN proposed in Kozachkov et al. (2022a) has biological motivations and encodes 1-contracting dynamics. This implies that all possible trajectories will exponentially converge, making the model robust to input disturbances.

Whilst the models above address a variety of problems, each model is quite limited in the range of dynamics they can learn, which is a problem when trying to design a generalised model for learning time-invariant dynamical systems. With respect to (2), the Lurie network has more flexibility than all of these models. Firstly, it includes all three weight matrices $(A, B, C)$ and biases $(b_x, b_y)$, whereas the models mentioned above fix at least one of these parameters. Secondly, the constraints imposed, and the corresponding parametrisations, allow the model to learn a variety of dynamics including, but not limited to, those mentioned above. The only limitation is that the dynamics must converge in some way. This includes certain types of chaotic systems, for example Thomas' cyclically symmetric attractor Thomas (1999). At the edge of the chaotic regime, the trajectories converge to a strange attractor, which could be modelled by a 3-contracting Lurie network.

The Lurie network is also related to a class of feed-forward models named *implicit* or *equilibrium networks*. These models use an implicit equation to express the relationship between the model output, layer outputs and model input in a compact vectorised form El Ghaoui et al. (2021). Like the Lurie network, these models can be represented by the interconnection of a linear time-invariant system and a nonlinearity. This makes analysis tools from control theory, such as Lipschitz bounds, applicable to these models Fazlyab et al. (2019). An additional connection is that the solution to the implicit equations correspond to equilibrium points of a Lurie system Revay et al. (2020).

As mentioned in the introduction, the $k$-contraction constraints used in this paper have an interesting connection to some properties observed in biological learning systems. A 2-contracting model can replicate the behaviour of associative memory, where every stored pattern corresponds to an equilibrium Kozachkov et al. (2023). Furthermore, a 3-contracting model can replicate the dynamics of working memory, where patterns are retained as attractor states Kozachkov et al. (2022b). Due to the 2–contraction (3-contraction) constraints, the equilibrium points (attractor states) must lie on a line (plane). As a result, the conditions developed in this paper could be of interest to neuroscientists and ML researchers interested in memory storage and retrieval Ramsauer et al. (2020); Hopfield (1984); Krotov & Hopfield (2020). The formation of the GLN also has connections to biology. For example, constructing larger systems from smaller modules can be motivated by evolutionary biology Simon (1962) where the name *facilitated variation* Gerhart & Kirschner (2007) is used to describe the development of traits in response to adaptation of the regulatory elements that connect modules, rather than the core components themselves. This was investigated in Kozachkov et al. (2022a) where the weights of the individual RNNs satisfied a 1-contraction condition but were fixed. Only the graph coupling weights were learnt during training.

The relationship between properties guaranteed by $k$-contraction analysis and those observed in associative and working memory suggest the $k$-contracting Lurie network possesses a number of appealing properties for an ML model; hence, it may be suitable for a wider class of ML problems. This proposition is supported by the successful application of the stability-constrained RNN models from §3.1 (special cases of a Lurie network) on a wide range of ML tasks. Since the Lurie network is a more structured, time-invariant example

of a neural ODE Chen et al. (2018), it will also be applicable to a similar array of tasks. Beyond modelling time-invariant dynamical systems, this includes image classification and continuous normalising flows Kidger (2022). The only limiting requirement is that the input must be passed in through the initial condition which in some cases, such as image classification, may require a pre-processing layer.

## 7 Conclusion

The Lurie network was presented as a novel and unifying time-invariant neural ODE. Stability results were presented which verified a generalised form of stability of the Lurie network, namely $k$-contraction. These results were incorporated in a principled approach for constructing $k$-contracting Lurie networks and graph Lurie networks which, unlike existing stability-constrained models, could be trained to model multi-stable and orbitally stable systems, such as those observed in associative and working memory. Empirical results showed the benefit of the graph structure and the $k$-contraction constraints through improved prediction accuracy, out of distribution generalisation and robustness on a range of simulated dynamical systems. These same models also compared favourably against well known stability-constrained models, which have been shown to be special cases of the Lurie network, and an unconstrained neural ODE. Future theoretical work will try to expand the class of systems the $k$-contracting Lurie network can be optimised over, by obtaining similar results for systems which are $k$-contracting in a diagonal metric. It would also be interesting to find conditions on the inputs of a time-varying Lurie network for which the convergence properties of the $k$-contracting Lurie network are also upheld, making the Lurie network applicable to sequential processing tasks. Finally, due to the connections with biological neural processes, we would like to empirically investigate the performance of the model on a wider range of machine learning tasks. The recently proposed working memory benchmark Sikarwar & Zhang (2024) is of particular interest due to its relationship with 3-contracting dynamics.

### Acknowledgments

This work was supported in part by the Defence Science and Technology Laboratory (DSTL) and the UK Research and Innovation (UKRI) Centre of Machine Intelligence for Nano-electronic Devices and Systems [EP/S024298/1].

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

## A  Extended Preliminaries

As many of the tools used in this paper are not well-known in the machine learning community, a condensed background is presented here, based on results from Bar-Shalom et al. (2023); Wu et al. (2022). The first section is on compound matrices, an important algebraic tool needed for generalising contraction analysis to $k$-contraction analysis. Following that, a geometric interpretation of the $k$-compound matrix is stated through its relationship to the volume of a parametrised set. Finally, the results in these two sections are utilised to define and provide intuition for the convergence of $k$-contracting dynamics.

### A.1  Compound Matrices

In this section, we document several known definitions and algebraic results related to compound matrices. The results are included without proof; the interested reader should refer to Bar-Shalom et al. (2023) for a more detailed tutorial on the topic.

Let $n$ be a positive integer and fix $k \in [1, n]$. The ordered set of *increasing* sequences of $k$ integers from $[1, n]$ is denoted by $Q(k, n)$. For example: $Q(3, 4) = \{(1, 2, 3), (1, 2, 4), (1, 3, 4), (2, 3, 4)\}$.

Now consider a matrix $W \in \Re^{n \times m}$. For $\alpha \in Q(k, n)$ and $\beta \in Q(k, m)$, the matrix $W[\alpha|\beta]$ denotes the $k \times k$ sub-matrix obtained by taking the entries of $W$ along the rows indexed by $\alpha$ and columns indexed by $\beta$. As an example, if $k = 2$ and $n = m = 4$, then $Q(2, 4) = \{(1, 2), (1, 3), (1, 4), (2, 3), (2, 4), (3, 4)\}$. The sub-matrix $W[(1, 2)|(3, 4)]$ would then be given by

$$W[(1, 2)|(3, 4)] = \begin{bmatrix} w_{13} & w_{14} \\ w_{23} & w_{24} \end{bmatrix}$$

The k-minors of the matrix $W$ are defined as $W(\alpha|\beta) := \det(W[\alpha|\beta])$.

**Definition 1 ($k$-multiplicative compound)** *Let* $W \in \Re^{n \times m}$ *and fix* $k \in [1, \min(n, m)]$. *The $k$-multiplicative compound of* $W$, *denoted* $W^{(k)}$, *is the* $\binom{n}{k} \times \binom{m}{k}$ *matrix containing all the $k$-minors of* $W$ *ordered lexicographically.*

For example, if we have $n = m = 3$ and $k = 2$ then $\alpha, \beta \in Q(2, 3) = \{(1, 2), (1, 3), (2, 3)\}$ and

$$W^{(2)} = \begin{bmatrix} W\big((1,2)|(1,2)\big) & W\big((1,2)|(1,3)\big) & W\big((1,2)|(2,3)\big) \\ W\big((1,3)|(1,2)\big) & W\big((1,3)|(1,3)\big) & W\big((1,3)|(2,3)\big) \\ W\big((2,3)|(1,2)\big) & W\big((2,3)|(1,3)\big) & W\big((2,3)|(2,3)\big) \end{bmatrix}$$

Some important special cases include

$$W^{(1)} = W \qquad W^{(n)} = \det(W) \qquad (pI_n)^{(k)} = p^k I_s \qquad W \in \mathcal{D}^n \to W^{(k)} \in \mathcal{D}^s \qquad (13)$$

with $s := \binom{n}{k}$. Next, we present a series of algebraic results concerned with the $k$-multiplicative compound.

**Fact 1 (Cauchy-Binet Formula)** *If* $U \in \Re^{n \times m}$, $V \in \Re^{m \times p}$ *and* $k \in [1, \min(n, m, p)]$, *then*

$$(UV)^{(k)} = U^{(k)} V^{(k)}$$

**Fact 2** *Fix* $k \in [1, \min(n, m)]$. *As a consequence of Definition 1, if* $W \in \Re^{n \times m}$ *then*

$$(W^\top)^{(k)} = (W^{(k)})^\top$$

**Fact 3** *Fix* $k \in [1, n]$. *If* $W \in \Re^{n \times n}$ *is non-singular, then by Theorem 1*

$$(W^{-1})^{(k)} = (W^{(k)})^{-1}$$

**Fact 4** *Fix* $k \in [1, \min(n, m, p)]$. *If* $W \in \Re^{n \times n}$, $U \in \Re^{p \times n}$ *and* $V \in \Re^{n \times p}$, *then by Theorem 1*

$$(UWV)^{(k)} = U^{(k)} W^{(k)} V^{(k)}$$

**Fact 5** *Fix $k \in [1, n]$. An implication of Theorem 1 is that if $W \in \Re^{n \times n}$ with eigenvalues $\lambda_1, \ldots, \lambda_n$, then the eigenvalues of $W^{(k)}$ are the $\binom{n}{k}$ products*

$$\Big\{ \prod_{l=1}^{k} \lambda_{i_l} : 1 \le i_1 < \cdots < i_k \le n \Big\}$$

We now introduce the definition of a second compound matrix, the $k$-additive compound, and a set of algebraic results related to it.

**Definition 2 ($k$-additive compound)** *Let $W \in \Re^{n \times n}$ and $k \in [1, n]$. The $k$-additive compound of $W$ is the $\binom{n}{k} \times \binom{n}{k}$ matrix defined by*

$$W^{[k]} := \frac{d}{d\epsilon} \big( I_n + \epsilon W \big)^{(k)} \big|_{\epsilon=0}$$

Special cases include

$$W^{[1]} = W \qquad\qquad W^{[n]} = tr(W) \qquad\qquad (pI_n)^{[k]} = kpI_s \qquad\qquad W \in \mathcal{D}^n \to W^{[k]} \in \mathcal{D}^s \qquad (14)$$

with $s := \binom{n}{k}$. Like before, we now present some useful algebraic results related to the $k$-additive compound.

**Fact 6** *If $W \in \Re^{n \times n}$ and $k \in [1, n]$, then as a consequence of Definition 2*

$$(W^\top)^{[k]} = (W^{[k]})^\top$$

**Fact 7** *Fix $k \in [1, n]$. For $W \in \Re^{n \times n}$ with eigenvalues $\lambda_1, \ldots, \lambda_n$, the eigenvalues of $W^{[k]}$ are the $\binom{n}{k}$ sums*

$$\{ \sum_{l=1}^{k} \lambda_{i_l} : 1 \le i_1 < \cdots < i_k \le n \}$$

An important consequence of Fact 7 is that if $W$ is positive definite (semi-definite), then this property is upheld by $W^{[k]}$. Opposite conclusions can be drawn if $W$ is negative definite (semi-definite).

**Fact 8** *Fix $k \in [1, n]$. If $U, V \in \Re^{n \times n}$, then*

$$(U + V)^{[k]} = U^{[k]} + V^{[k]}$$

**Fact 9** *Fix $k \in [1, \min(n, p)]$. If $W \in \Re^{n \times n}$, $U \in \Re^{p \times n}$, $V \in \Re^{n \times p}$ and $UV = I_p$, then*

$$(UWV)^{[k]} = U^{(k)} W^{[k]} (U^{(k)})^{-1}$$

## A.2 Volume of $k$-sets

This section aims to provide a clear geometric interpretation of the $k$-multiplicative compound. The section begins by defining a $k$-set (the codomain of a function dependent on $k$ variables) before presenting Theorem 8, the key result which exposes the relationship between the volume of a $k$-set and the $k$-multiplicative compound of the Jacobian. A $k$-parallelotope is then shown as an example. Like before, these are existing results, so are presented without proof. Refer to Wu et al. (2022) for more information.

**Definition 3 ($k$-sets)** *Consider a compact set $\mathcal{D} \subset \Re^k$ and a continuous differentiable map $\Psi : \mathcal{D} \to \Re^n$, with $k \in [1, n]$. The codomain of $\Psi$ is given by the parametrised set*

$$\Psi(D) := \{ \Psi(r) : r \in \mathcal{D} \} \subseteq \Re^n \qquad (15)$$

*Since $\mathcal{D}$ is compact and $\Psi(\cdot)$ is continuous, $\Psi(\mathcal{D})$ is a closed set.*

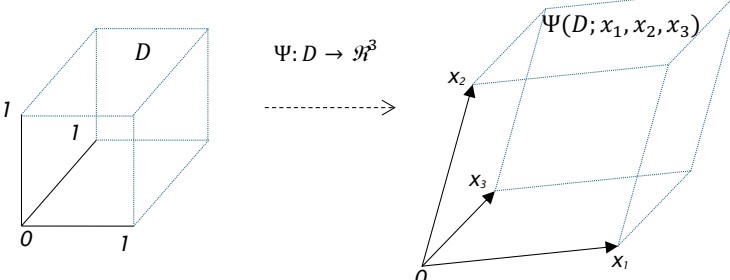

Figure 2: The 3-parallelotope with vertices $x_1, x_2, x_3 \in \Re^3$ parametrised by the unit cube, $\mathcal{D}$.

**Theorem 8 (Volume of $k$-sets)** *Fix $k \in [1, n]$. Consider a compact set $\mathcal{D} \subset \Re^k$ and a continuously differentiable map $\Psi : \mathcal{D} \to \Re^n$. The volume of the parametrised set (15) is given by*

$$vol\left(\Psi(D)\right) = \int_{\mathcal{D}} \left| J_{\Psi}^{(k)}(r) \right| dr$$

*where $J_{\Psi}(r) = \left[ \frac{\partial \Psi(r)}{\partial r_1} \quad \ldots \quad \frac{\partial \Psi(r)}{\partial r_k} \right]$ is the Jacobian of $\Psi$. Note that as $J_{\Psi} : \mathcal{D} \to \Re^{n \times k}$, it implies $J_{\Psi}^{(k)}(r) : \mathcal{D} \to \Re^s$ with $s = \binom{n}{k}$.*

Theorem 8 states that the volume of a $k$-set is governed by the $k$-multiplicative compound of the Jacobian. An important geometrical feature, is that for $k \in \{1, 2, 3\}$ the volume of the $k$-set is equivalent to the standard notions of length, area and volume. We now consider the $k$-parallelotope as an example of a $k$-set.

**Definition 4 ($k$-parallelotope)** *Fix $k \in [1, n]$ and let vectors $x_1, \ldots, x_k \in \Re^n$. The parallelotope generated by these vectors (and the zero vertex) is the set given by*

$$P(x_1, \ldots, x_k) := \left\{ \sum_{i=1}^{k} r_i x_i : r_i \in [0, 1] \, \forall \, i \right\}$$

Based on Definition 4, the $k$-parallelotope is a $k$-set with the following compact domain $\mathcal{D}$ and continuous differentiable function $\Psi(r; x_1, \ldots, x_k)$, as illustrated for $k = n = 3$ in Figure 2.

$$\mathcal{D} := \left\{ r \in \Re^k : r_i \in [0, 1] \, \forall \, i \in [1, k] \right\} \qquad\qquad \Psi(r; x_1, \ldots, x_k) := \sum_{i=1}^{k} r_i x_i$$

The Jacobian of the $k$-parallelotope is $J_{\Psi}(r) = X$ and from Theorem 8, the volume of the $k$-parallelotope is given by $|X^{(k)}|$.

### A.3  $k$-contraction Analysis

Consider the time-varying nonlinear system

$$\dot{x} = f(t, x) \tag{16}$$

where $f : \Re_+ \times \Re^n \to \Re^n$. It is assumed throughout that $f$ is continuously differentiable w.r.t $x$. Fix $k \in [1, n]$ and let $\mathcal{S}^k$ denote the unit simplex.

$$\mathcal{S}^k := \{ r \in \Re^k : r_i \geq 0 \text{ and } r_1 + \cdots + r_k \leq 1 \}$$

The convex combination of a set of initial conditions $x_1, \ldots, x_{k+1} \in \Re^n$ is defined by $h : \mathcal{S}^k \to \Re^n$.

$$h(r; x_1, \ldots, x_{k+1}) := \sum_{i=1}^{k} r_i x_i + \left( 1 - \sum_{i=1}^{k} r_i \right) x_{k+1}$$

The set $h(\mathcal{S}^k)$ is a $k$-set and can be thought of as a $k$-dimensional body of states representing initial conditions of (16). We now define $w_i(t, r)$ as a measure of the sensitivity of a solution to (16) at time $t$, to a change in the initial condition $h(r)$, caused by a change in $r_i$.

$$w_i(t, r) := \frac{\partial x(t, h(r))}{\partial r_i} \quad \text{where } w_i(0, r) = \frac{\partial h(r)}{\partial r_i} = x_i - x_{k+1} \quad \text{for all } i \in [1, k]$$

We are now ready to present the definition of $k$-contraction, followed by its geometric interpretation.

**Definition 5 ($k$-contraction)** *Fix $k \in [1, n]$. The nonlinear system (16) is $k$-contracting if there exists an $\eta > 0$ and a vector norm $|\cdot|$ such that for any $x_1, \ldots, x_{k+1} \in \Re^n$ and any $r \in \mathcal{S}^k$, the mapping $W : \Re_+ \times \mathcal{S}^k \to \Re^{n \times k}$ defined by $W(t, r) := \begin{bmatrix} w_1(t, r) & \ldots & w_k(t, r) \end{bmatrix}$ satisfies*

$$|W^{(k)}(t, r)| \leq \exp(-\eta t)|W^{(k)}(0, r)| \quad \forall\, t \in \Re_+$$

To explain the geometric meaning of this definition, pick a domain $\mathcal{D} \subseteq \mathcal{S}^k$ and recall that $h(\mathcal{D})$ is a $k$-set representing $k$-dimensional bodies of initial conditions for (16); thus, $x(t, h(\mathcal{D})) := \{x(t, h(r)) : r \in \mathcal{D}\}$ is a $k$-set describing how $k$-dimensional bodies evolve over time. We now leverage Theorem 8, to show how the volume of these bodies evolves over time when governed by (16).

$$\begin{aligned}
vol\left(x\big(t, h(\mathcal{D})\big)\right) &= \int_{\mathcal{D}} \left| J_x\big(t, h(r)\big)^{(k)} \right| dr \\
&= \int_{\mathcal{D}} \left| \left[ \frac{\partial x\big(t, h(r)\big)}{\partial r_1} \quad \ldots \quad \frac{\partial x\big(t, h(r)\big)}{\partial r_k} \right]^{(k)} \right| dr \\
&= \int_{\mathcal{D}} \left| W^{(k)}(t, r) \right| dr
\end{aligned}$$

When (16) is $k$-contracting, the volume of these bodies is upper bounded by the initial volume scaled by an exponentially decaying term.

$$\begin{aligned}
vol\left(x\big(t, h(\mathcal{D})\big)\right) &\leq \exp(-\eta t) \int_{\mathcal{D}} |W^{(k)}(0, r)| dr \\
&= \exp(-\eta t) \left| \left[ (x_1 - x_{k+1}) \quad \ldots \quad (x_k - x_{k+1}) \right]^{(k)} \right| \int_{\mathcal{D}} dr
\end{aligned}$$

Therefore, $k$-contraction of (16) implies the volume of $k$-dimensional bodies $x(t, h(\mathcal{D}))$ converges to zero at an exponential rate. This can also be interpreted as *the volume of $k$-dimensional bodies is contracting or converging to a $(k-1)$-dimensional subspace.* Figure 1 in the main paper gives an illustration of $(1, 2, 3)$-contracting systems where the 1-contracting system converges to a point, the 2-contracting system converges to a line and the 3-contracting system converges to a plane.

Many existing $k$-contraction results, including Theorem 1, are expressed in terms of matrix measures. An overview of their definitions and properties can be found in (Vidyasagar, 2002, Section 2.2). Theorem 1 provides a sufficient condition for verifying $k$-contraction in the 2-norm w.r.t a metric $P$. The 2-norm was chosen for this work due to its relationship with the eigenvalues of its argument, but other norms could be chosen. Furthermore, one may apply an invertible linear transformation $\Theta$ to $w_i(t, r)$ and the $k$-contraction analysis may be performed in this new domain whilst implying the same property holds in the original domain. This idea is made clear in Lohmiller & Slotine (1998) and translates analogously to the $k$-contraction case. When using such an invertible transformation, the system is said to be $k$-contracting w.r.t the metric $P = \Theta^\top \Theta$.

## B   Lurie Network

### B.1   Relationship Between Lurie Networks and Deep Feedforward Models

Consider the vector field $\dot{z} = f(z)$ defined by the following $L$-layer feedforward network

$$
\begin{aligned}
\dot{z} &= W_L \Phi(u_{L-1}) + b_L \\
u_{L-1} &= W_{L-1} \Phi(u_{L-2}) + b_{L-1} \\
u_{L-2} &= W_{L-2} \Phi(u_{L-3}) + b_{L-2} \\
&\vdots \\
u_2 &= W_2 \Phi(u_1) + b_2 \\
u_1 &= W_1 z + b_1
\end{aligned}
\tag{17}
$$

To illustrate the superior expressivity of the Lurie network, we wish to show that a special case of (2) can approximate the deep feedforward network (17). An alternative expression for (17) is

$$
\begin{aligned}
\dot{z} &= 0z + W_L \Phi(u_{L-1}) + b_L \\
\epsilon \dot{u}_{L-1} &= -u_{L-1} + W_{L-1} \Phi(u_{L-2}) + b_{L-1} \\
\epsilon \dot{u}_{L-2} &= -u_{L-2} + W_{L-2} \Phi(u_{L-3}) + b_{L-2} \\
&\vdots \\
\epsilon \dot{u}_2 &= -u_2 + W_2 \Phi(u_1) + b_2 \\
u_1 &= W_1 z + b_1
\end{aligned}
\tag{18}
$$

where $\epsilon \to 0$. Defining a new state $x := \begin{bmatrix} z & u_{L-1} & u_{L-2} & \dots & u_3 & u_2 \end{bmatrix}^\top$ and an output vector $y := \begin{bmatrix} u_{L-1} & u_{L-2} & \dots & u_2 & u_1 \end{bmatrix}$ it is clear that (18) is a special case of the Lurie network (2) with state-space matrices and biases defined by the sparse structures below.

$$
A = \epsilon^{-1} \begin{bmatrix}
0 & 0 & 0 & \dots & 0 & 0 \\
0 & -I & 0 & \dots & 0 & 0 \\
0 & 0 & -I & \dots & 0 & 0 \\
\vdots & \vdots & \vdots & \ddots & \vdots & \vdots \\
0 & 0 & 0 & \dots & & -I
\end{bmatrix}
\qquad
B = \epsilon^{-1} \begin{bmatrix}
0 & W_L & 0 & \dots & 0 & 0 \\
0 & 0 & W_{L-1} & \dots & 0 & 0 \\
0 & 0 & 0 & \dots & 0 & 0 \\
\vdots & \vdots & \vdots & \ddots & \vdots & \vdots \\
0 & 0 & 0 & \dots & 0 & W_2
\end{bmatrix}
$$

$$
C = \begin{bmatrix}
0 & I & 0 & \dots & 0 & 0 \\
0 & 0 & I & \dots & 0 & 0 \\
0 & 0 & 0 & \dots & 0 & 0 \\
\vdots & \vdots & \vdots & \ddots & \vdots & \vdots \\
W_1 & 0 & 0 & \dots & 0 & 0
\end{bmatrix}
\qquad
b_x = \epsilon^{-1} \begin{bmatrix}
b_L \\
b_{L-1} \\
b_{L-2} \\
\vdots \\
b_2
\end{bmatrix}
\qquad
b_y = \begin{bmatrix}
0 \\
0 \\
0 \\
\vdots \\
b_1
\end{bmatrix}
$$

This is just one realisation of a Lurie network which approximates a feedforward network. Other permutations of the state would result in different realisations of the Lurie networks weights and biases. Finally, due to the division by $\epsilon$, it would not be possible to train a Lurie network to have the exact form of a feedforward network; however, this analysis shows that it is possible to approximate the strucutre of a deep feedforward network with a Lurie network.

### B.2 Further Examples

**Neural Oscillators:** This example is from the graph ML literature Lanthaler et al. (2024). The state of the general neural oscillator is governed by a second order ODE; however, it's equivalent first order representation takes the form (2) with one possible realisation given by $C_{21} \in \Re^{n \times n}$, $b_x = 0$ and

$$A = \begin{bmatrix} 0 & I \\ 0 & 0 \end{bmatrix} \qquad B = \begin{bmatrix} 0 & 0 \\ 0 & I \end{bmatrix} \qquad C = \begin{bmatrix} 0 & 0 \\ C_{21} & 0 \end{bmatrix} \qquad b_y = \begin{bmatrix} 0 \\ b_{y2} \end{bmatrix}$$

The solution is then passed through an affine readout layer.

**Graph Coupled Oscillators:** Another example of a second order ODE from the graph ML literature Rusch et al. (2022). The state is defined as a matrix, but this can simply be recast in a vectorised form which relates to (2) if a linear coupling function is chosen. This requires the weights to have a block matrix form, where one possible realisation is defined by $C_{21} \in \Re^{n \times n}$, $b_x = b_y = 0$ and

$$A = \begin{bmatrix} 0 & I \\ -\gamma I & -\alpha I \end{bmatrix} \qquad B = \begin{bmatrix} 0 & 0 \\ 0 & I \end{bmatrix} \qquad C = \begin{bmatrix} 0 & 0 \\ C_{21} & 0 \end{bmatrix}$$

**LSSM:** When the external input is replaced by nonlinear output feedback (i.e., $u(t) \equiv \Phi(y)$) and $D = 0$, the linear state-space layer used in S4 Gu et al. (2021) and Hippo Gu et al. (2020) is related to (2) with $A$ being a lower triangular Hippo matrix , $B \in \Re^{n \times m}$, $C \in \Re^{m \times n}$ and $b_x = b_y = 0$.

## C Proofs

### C.1 Proof of Theorem 2

We aim to verify Theorem 1 for the particular case where the nonlinear system is described by the Lurie network (2). Our proof begins with Theorem 9, which restates (Ofir et al., 2024, Theorem 1). This result is sufficient to satisfy Theorem 1 for systems of the form (19).

**Theorem 9** *Fix $k \in [1, n]$ and consider the system below.*

$$\dot{x} = \bar{A}x(t) - \bar{B}\Psi(t, y) \qquad\qquad y = \bar{C}x \qquad (19)$$

*If there exists $\eta_1, \eta_2 > 0$ and an invertible $\Theta \in \Re^{n \times n}$ such that*

$$P^{(k)}\bar{A}^{[k]} + (\bar{A}^{[k]})^\top P^{(k)} + \Theta^{(k)}\Big((\Theta\bar{B}\bar{B}^\top\Theta)^{[k]} + (\Theta^{-1}\bar{C}^\top\bar{C}\Theta^{-1})^{[k]}\Big)\Theta^{(k)} \preceq -\eta_1 P^{(k)} \qquad (20)$$

*and*

$$\Big(\Theta^{-1}\bar{C}^\top(J_\Psi^\top(t, y)J_\Psi(t, y) - I_m)\bar{C}\Theta^{-1}\Big)^{[k]} \preceq -\eta_2 I_s \quad \forall\, t \in \Re_+ \text{ and } y \in \Re^m \qquad (21)$$

*where $s = \binom{n}{k}$, then (19) is $k$-contracting in the $2$-norm w.r.t the metric $P := \Theta^\top\Theta$.*

We first need to express the Lurie network in the form (19). By (3) there exists $\gamma < 0$ satisfying

$$0 < \gamma^2 < \alpha_k^2 \quad\text{and}\quad g^2\sum_{i=1}^k \sigma_i^2(B)\sigma_i^2(C) < \gamma^2 k \qquad (22)$$

Using $\gamma$, we can express (2) in the form (19) through the definitions below, where the dependence on $t$ has been dropped from $\Psi$.

$$\bar{A} := A \qquad\qquad \bar{B} := \gamma I_n \qquad\qquad \bar{C} := I_n \qquad\qquad \Psi(x) := -\gamma^{-1}B\Phi(Cx + b_y) - \gamma^{-1}b_x \qquad (23)$$

The next step is to verify (20). Subbing (23) into the left hand side of (20) and assuming $\Theta = \Theta^\top$ results in the first equality. Setting $P := pI_n$ with $p > 0$ results in the second. Now we must leverage some of the facts presented in §A.1. Using the relevant special cases from (13) and (14) leads to equality three and consequently applying Fact 6 and Fact 8 results in equality four. Re-applying (14) and Fact 8 results in the final equality.

$$\begin{aligned}
&= P^{(k)}A^{[k]} + (A^{[k]})^\top P^{(k)} + \Theta^{(k)}\Big((\gamma^2 P)^{[k]} + (P^{-1})^{[k]}\Big)\Theta^{(k)} \\
&= (pI_n)^{(k)}A^{[k]} + (A^{[k]})^\top(pI_n)^{(k)} + (p^{\frac{1}{2}}I_n)^{(k)}\Big((\gamma^2 pI_n)^{[k]} + (p^{-1}I_n)^{[k]}\Big)(p^{\frac{1}{2}}I_n)^{(k)} \\
&= p^k\big(A^{[k]} + (A^{[k]})^\top\big) + k(\gamma^2 p + p^{-1})p^k I_s \\
&= p^k\big((A + A^\top)^{[k]} + k(\gamma^2 p + p^{-1})I_s\big) \\
&= p^k\big(A + A^\top + (\gamma^2 p + p^{-1})I_n\big)^{[k]}
\end{aligned}$$

If the matrix above is negative definite, then (20) is satisfied for some suitably chosen $\eta_1 > 0$. This is true when

$$\big(A + A^\top + (\gamma^2 p + p^{-1})I_n\big)^{[k]} \prec 0$$

By Fact 7, the inequality above can be equivalently expressed as a condition on the sum of the $k$ largest eigenvalues of the matrix inside the $k$-compound operator. Leveraging (Petersen et al., 2008, Eq. 285) allows us to separate $p$ from the eigenvalues of the symmetric component of $A$, resulting in the equality below.

$$\sum_{i=1}^k \lambda_i\big(A + A^\top + (\gamma^2 p + p^{-1})I_n\big) = k(\gamma^2 p + p^{-1}) + \sum_{i=1}^k \lambda_i(A + A^\top) < 0$$

By the definition of $\alpha_k$ in Theorem 2, this simplifies to

$$\gamma^2 p^2 + 2\alpha_k p + 1 < 0$$

For $\gamma$ satisfying (22), the quadratic inequality always emits at least one solution $p = -\alpha_k^{-1}$ .

The final step is to verify (21). The Jacobian of $\Psi$, as defined in (23), is

$$J_\Psi(x) = -\gamma^{-1} B J_\Phi C$$

For $\Theta = p^{\frac{1}{2}} I_n$ and the definitions from (23), the left hand side of (21) reduces to

$$= \left(p^{-1} J_\Psi^\top J_\Psi - p^{-1} I_n\right)^{[k]}$$

If the matrix above is negative definite, then (21) is satisfied for some suitably chosen $\eta_2 > 0$. Repeating the same steps as above, this negative definite requirement reduces to the inequality below.

$$\sum_{i=1}^k \lambda_i(p^{-1} J_\Psi^\top J_\Psi) - kp^{-1} = p^{-1} \sum_{i=1}^k \sigma_i^2(J_\Psi) - kp^{-1} < 0$$

Subbing in the definition of $J_\Psi$ and applying the well-known property of singular values (Horn & Johnson, 1994, Theorem 3.3.14), then (21) is true if

$$\gamma^{-2} \sum_{i=1}^k \sigma_i^2(B)\sigma_i^2(J_\Phi)\sigma_i^2(C) < k$$

By the assumption made on the slope of $\Phi$, this inequality will always be satisfied if (3) holds. □

## C.2 Proof of Theorem 3

For this proof, we aim to directly verify Theorem 1 for $\Theta \in \mathcal{D}^n$ where $f$ represents the Lurie network (2). We start by substituting the Jacobian of the Lurie network into the left hand side of (1), followed by the application of Fact 8 to obtain the second equality. The subadditivity property of the matrix measure $\mu_2$ was then leveraged to split the terms (Vidyasagar, 2002, Section 2.2). As the second term is difficult to manipulate, we rely on the simplifying assumption $C = B^{-1}$ in order to apply Fact 9. As $\Theta, B, J_\Phi \in \mathcal{D}^n$, so are both of their $k$-compound counterparts (13) (14), which means the $\Theta, B$ terms cancel out. We then apply the property $\mu_2(\cdot) \leq ||\cdot||_2$ (Vidyasagar, 2002, Theorem 16) and Fact 7, which allows us to leverage the slope restricted assumption on $\Phi$. By the relevant special case of the $k$-additive compound (14), we can directly calculate the 2-norm. Finally, $kg$ can be incorporated in $\mu_2$ as shown in the final inequality.

$$\begin{aligned}
\mu_{2,\Theta^{(k)}}(J_f^{[k]}(t,x)) &= \mu_{2,\Theta^{(k)}}\left((A + BJ_\Phi C)^{[k]}\right) \\
&= \mu_{2,\Theta^{(k)}}\left(A^{[k]} + (BJ_\Phi(y)C)^{[k]}\right) \\
&\leq \mu_{2,\Theta^{(k)}}(A^{[k]}) + \mu_{2,\Theta^{(k)}}\left((BJ_\Phi C)^{[k]}\right) \\
&= \mu_{2,\Theta^{(k)}}(A^{[k]}) + \mu_2(\Theta^{(k)} B^{(k)} J_\Phi^{[k]} B^{-(k)} \Theta^{-(k)}) \\
&= \mu_{2,\Theta^{(k)}}(A^{[k]}) + \mu_2(J_\Phi^{[k]}) \\
&\leq \mu_{2,\Theta^{(k)}}(A^{[k]}) + ||(gI_n)^{[k]}||_2 \\
&= \mu_{2,\Theta^{(k)}}(A^{[k]}) + kg \\
&= \mu_2(\Theta^{(k)} A^{[k]} \Theta^{-(k)} + kgI_n)
\end{aligned}$$

If the final inequality is negative, then Theorem 1 will be satisfied for some suitably chosen $\eta$. This is equivalent to the matrix inequality below.

$$\frac{1}{2}\left(\Theta^{(k)} A^{[k]} \Theta^{-(k)} + \Theta^{-(k)}(A^{[k]})^\top \Theta^{(k)} + 2kgI_n\right) \prec 0$$

Multiplying on the left by $2\Theta^{(k)}$ and on the right by $\Theta^{(k)}$ results in (4). □

### C.3 Proof of Theorem 4

The aim of this proof is to verify Theorem 1 when $f$ is the GLN (6). We begin by expressing the Jacobian as a sum of the Jacobian of the $q$ independent Lurie networks, $J_{\text{indep}}$, and Jacobian of the coupling term $J_{\text{couple}}$

$$J_f(x) = J_{\text{indep}}(x) + J_{\text{couple}}(x)$$

where $J_{\text{indep}}(x) = A_G + B_G J_\Phi C_G$ and $J_{\text{couple}}(x) = L$. Subbing $J_f$ into the left hand side of (1) and applying the subadditivity property of $\mu_2$ results in

$$\mu_{2,\Theta^{(k)}}(J_f^{[k]}) \leq \mu_{2,\Theta^{(k)}}(J_{\text{indep}}^{[k]}) + \mu_{2,\Theta^{(k)}}(J_{\text{couple}}^{[k]})$$

As we assume the $q$ independent Lurie networks are $k$-contracting in the 2-norm w.r.t the metric $P = \text{blockdiag}(P_1, \ldots, P_q)$, we know that $\mu_{2,\Theta^{(k)}}(J_{\text{indep}}^{[k]}) < 0$. Under this assumption, Theorem 1 is satisfied if

$$\mu_{2,\Theta^{(k)}}(J_{\text{couple}}^{[k]}) \leq 0$$

This is equivalent to the matrix inequality below, where $J_{\text{couple}}$ has been subbed in.

$$\frac{1}{2}\left(\Theta^{(k)} L^{[k]} \Theta^{-(k)} + \Theta^{-(k)}(L^{[k]})^\top \Theta^{(k)}\right) \preceq 0$$

Multiplying on the left by $2\Theta^{(k)}$ and on the right by $\Theta^{(k)}$ results in (7). $\qquad\square$

### C.4 Proof of Theorem 5 and Theorem 6

The aim of this proof is to express the weights of the Lurie network (2) such that Theorem 2 is always satisfied. More formally, this requires $A, B, C \in \Omega_2(g, k)$ to always hold. As both theorems share the same proof until the final step, where $\Sigma_A$ is defined, they have been combined into one proof.

To expose the singular values of $B, C$, we leverage the singular value decomposition, as in (9b) and (10a). This requires the matrices $U_B, U_C, V_B, V_C$ to be orthogonal. We can immediately use the unconstrained parametrisation of the orthogonal class from Lezcano-Casado & Martınez-Rubio (2019) to express these matrices as unconstrained symmetric matrices. The matrices $\Sigma_B, \Sigma_C$ contain the singular values of the respective matrix, hence $\Sigma_B \in \mathcal{D}_+^{nm}$ and $\Sigma_C \in \mathcal{D}_+^{mn}$. We also treat these as unconstrained sets since any element can be obtained by taking the absolute value of an unconstrained diagonal matrix with the same shape.

To verify Theorem 2, we can combine $\alpha_k < 0$ and (3) into one inequality representing the intersection of the two sets.

$$\sum_{i=1}^{k} \lambda_i(A + A^\top) < -2k\sqrt{\frac{z_k}{k}} \quad \text{where } z_k := g^2 \sum_{i=1}^{k} \sigma_i^2(B)\sigma_i^2(C) \tag{24}$$

Thanks to the definition of $B$ and $C$, the right hand side is a function of the hyperparameters $g, k$ and elements of the parameters $\Sigma_B, \Sigma_C$, so can be easily computed using sort and sum functions.

To impose this constraint directly on the eigenvalues of the symmetric component of $A$, we express $A$ as a sum of symmetric and skew-symmetric matrices. The skew-symmetric matrix is unconstrained, so this can be left as it is. Finally, we apply the eigenvalue decomposition of a symmetric matrix to obtain (9a) (10a). This expression allows us to directly place the constraint above on the diagonal matrix $\Sigma_A$.

This is where Theorem 5 and Theorem 6 differ. Defining $\Sigma_A$ as in (9a) ensures

$$\lambda_i(A + A^\top) < -2\sqrt{\frac{z_k}{k}} \quad \text{for all } i \in [1, n]$$

which guarantees both conditions of Theorem 2 will be satisfied; however, conservatism is introduced as all the eigenvalues must be negative. Theorem 6 was established to address this issue. Defining $\Sigma_A$ as in (10b)

guarantees both conditions of Theorem 2 will be satisfied via (24). The definition of $\Sigma_A$ is split into one unconstrained block for the first $(k-1)$-eigenvalues, a block for $\lambda_k(A + A^\top)$ which is defined to ensure (24) holds, and finally a block for the remaining eigenvalues which must be defined to ensure the $k$ eigenvalues involved in (24) are the largest. $\qquad\square$

### C.5 Intersection Proof

The set $\Omega_2(g, k)$ is defined by (8a). For $k \in [2, n]$, the set of $(k-1)$-contracting systems is denoted by

$$\Omega_2(g, k-1) = \left\{ (A, B, C) \mid \alpha_{k-1} < 0 \,,\; z_{k-1} < \alpha_{k-1}^2(k-1) \right\}$$

**Theorem 10** *For $k \in \{2, 3\}$, the sets $\Omega_2(g, k-1)$ and $\Omega_2(g, k)$ intersect when*

$$g^2 \sigma_k^2(B)\sigma_k^2(C) < \frac{1}{4k}\lambda_k^2 + \frac{k-1}{k}\alpha_{k-1}\lambda_k - \frac{k-1}{k}\alpha_{k-1}^2 \tag{25}$$

*Proof*:

We first note the following useful relationships, where $\lambda_k := \lambda_k(A + A^\top)$.

$$\alpha_k = \frac{1}{2k}\lambda_k + \frac{k-1}{k}\alpha_{k-1} \qquad\qquad z_k = g^2\sigma_k^2(B)\sigma_k^2(C) + z_{k-1} \tag{26}$$

From (26) it is clear that $\alpha_k < 0$ is true when

$$\lambda_k < -2(k-1)\alpha_{k-1}$$

A subset of these solutions occurs when the first condition of $\Omega_2(g, k-1)$ holds. That is $\alpha_{k-1} < 0$.

From (26) it is also clear that $z_k < \alpha_k^2 k$ is true when

$$z_{k-1} < \alpha_k^2 k - g^2\sigma_k^2(B)\sigma_k^2(C)$$

If $\alpha_{k-1}^2(k-1) < \alpha_k^2 k - g^2\sigma_k^2(B)\sigma_k^2(C)$ then a subset of these solutions occurs when the second condition of $\Omega_2(g, k-1)$ holds. After rearranging, this is equivalent to

$$g^2\sigma_k^2(B)\sigma_k^2(C) < \alpha_k^2 k - \alpha_{k-1}^2(k-1)$$

The RHS can equivalently be expressed by a quadratic equation in $\lambda_k$

$$
\begin{aligned}
\alpha_k^2 k - \alpha_{k-1}^2(k-1) &= \left(\frac{1}{2k}\lambda_k + \frac{k-1}{k}\alpha_{k-1}\right)^2 k - \alpha_{k-1}^2(k-1) \\
&= \left(\frac{1}{4k^2}\lambda_k^2 + \frac{k-1}{k^2}\alpha_{k-1}\lambda_k + \frac{(k-1)^2}{k^2}\alpha_{k-1}^2\right)k - \alpha_{k-1}^2(k-1) \\
&= \frac{1}{4k}\lambda_k^2 + \frac{k-1}{k}\alpha_{k-1}\lambda_k + \frac{(k-1)^2}{k}\alpha_{k-1}^2 - \alpha_{k-1}^2(k-1) \\
&= \frac{1}{4k}\lambda_k^2 + \frac{k-1}{k}\alpha_{k-1}\lambda_k - \frac{k-1}{k}\alpha_{k-1}^2 \\
&:= Q_k(\lambda_k, \alpha_{k-1})
\end{aligned}
$$

Therefore, $\Omega_2(g, k-1)$ intersects with $\Omega_2(g, k)$ if (25) holds.

Finding the roots of $Q_k(\lambda_k, \alpha_{k-1})$ will allow us to check what intervals of $\lambda_k$ the $Q_k(\lambda_k, \alpha_{k-1}) > 0$. This is required since the LHS of (25) must be positive.

$$
\begin{aligned}
\lambda_k^r &= 2(1-k)\alpha_{k-1} \pm 2k\sqrt{\frac{(k-1)^2}{k^2}\alpha_{k-1}^2 + \frac{k-1}{k^2}\alpha_{k-1}^2} \\
&= 2(1-k)\alpha_{k-1} \pm 2\sqrt{k(k-1)}\alpha_{k-1} \\
&= 2\left(1-k \pm \sqrt{k(k-1)}\right)\alpha_{k-1}
\end{aligned}
$$

The roots for $k = 2$ and $k = 3$ are given by

$$
\lambda_2^r = (-2 \pm 2\sqrt{2})\alpha_1 \qquad\qquad\qquad \lambda_3^r = (-4 \pm 2\sqrt{6})\alpha_2
$$

Given $\alpha_{k-1} < 0$, we will test the intervals $\lambda_2 \in [-\infty, (-2+2\sqrt{2})\alpha_1]$ and $\lambda_3 \in [-\infty, (-4+2\sqrt{6})\alpha_2]$ to see if the co-domain of $Q_k(\lambda_k, \alpha_{k-1})$ is positive. These intervals ensure $\lambda_k < 0$ (a requirement for $\alpha_k < 0$ and $\alpha_{k-1} < 0$).

Using $\lambda_2 = \alpha_1$ and $\lambda_3 = \alpha_2$ as the test points, we see that

$$
Q_2(\alpha_1; \alpha_1) = \frac{1}{8}\alpha_1^2 + \frac{1}{2}\alpha_1^2 - \frac{1}{2}\alpha_1^2 \tag{27}
$$

$$
= \frac{1}{8}\alpha_1^2 > 0 \tag{28}
$$

$$
Q_3(\alpha_2; \alpha_2) = \frac{1}{12}\alpha_2^2 + \frac{2}{3}\alpha_2^2 - \frac{2}{3}\alpha_2^2 \tag{29}
$$

$$
= \frac{1}{12}\alpha_2^2 > 0 \tag{30}
$$

As a result, for $k \in \{2, 3\}$, the sets $\Omega_2(g, k-1)$ and $\Omega_2(g, k)$ intersect if $\lambda_k \in [-\infty, 2(1-k+\sqrt{k(k-1)})\alpha_{k-1}]$ and $g^2\sigma_k^2(B)\sigma_k^2(C) < Q_k(\lambda_k, \alpha_{k-1})$. If $(A, B, C) \in \Omega_2(g, 2)$ then $\lambda_2$ must be in this interval since $\lambda_2 \leq \lambda_1$. By the same logic, if $(A, B, C) \in \Omega_2(g, 3)$ then $\lambda_3$ must be in this interval since $\lambda_3 \leq \lambda_2 \leq \lambda_1$ .

Hence, the requirements for intersection can be simplified to just (25). $\qquad\square$

### C.6 Proof of Theorem 7

This proof aims to show that Theorem 4 is satisfied when $L$ is defined as in (12). First, multiply (7) on the left and right by $\Theta^{-(k)}$. Sequentially applying Fact 6, Fact 9 and Fact 8 results in

$$
\left(\Theta L \Theta^{-1} + \Theta^{-1}L^\top\Theta\right)^{[k]} \preceq 0
$$

Subbing in the definition of $L$ from (12) and recalling that $P = \Theta^\top\Theta$ leads to

$$
\left(G_{L2} + G_{L2}^\top\right)^{[k]} \preceq 0
$$

which is assumed to hold. $\qquad\square$

# D  Extended Empirical Evaluation

## D.1  Data

The data was synthetically generated by numerically integrating over the analytical models of each dynamical system. The integration was performed using the Euler method with step size $\delta = 1 \times 10^{-2}$.

### D.1.1  Opinion Dynamics

This model was presented in Ofir et al. (2024). It has the following state space equations

$$\dot{x} = -1.5I_3 + 0.5\Phi(Cx) + b$$

where

$$C = \begin{bmatrix} +1 & -1 & 0 \\ -1 & +1 & -1 \\ 0 & -1 & +1 \end{bmatrix} \qquad\qquad b = \begin{bmatrix} +0.2 \\ 0 \\ -0.2 \end{bmatrix}$$

and the tanh function is applied element-wise. The model is 1-contracting and has a unique equilibrium point at $b$.

### D.1.2  Hopfield Network

This model is a variation of the Hopfield network presented in Ofir et al. (2024). It has the following state space equations

$$\dot{x} = -2.5I_3 + B\Phi(x)$$

where

$$B = \begin{bmatrix} 1 & 1 & 1 \\ 1 & 1 & 1 \\ 1 & 1 & 1 \end{bmatrix}$$

and the tanh function is also applied element-wise. The model is 2-contracting and has two stable equilibrium points: $e_1 = \begin{bmatrix} 0.79 & 0.79 & 0.79 \end{bmatrix}^\top$, $e_2 = -e_1$; and an unstable equilibrium point $e_3 = 0$.

### D.1.3  Simple Attractor

This model was presented in Cecilia et al. (2023). It has the following state space equations

$$\dot{x} = Ax + B\Phi(Cx)$$

where

$$A = \begin{bmatrix} 0 & 1 & -2 \\ -1 & 0 & -1 \\ 0.5 & 0 & -0.5 \end{bmatrix} \qquad B = \begin{bmatrix} 0 & 0 & 0 \\ 0 & 0 & 0 \\ -0.5 & 0 & 0 \end{bmatrix} \qquad C = \begin{bmatrix} 0 & 0 & 0 \\ 0 & 0 & 0 \\ 1 & 0 & 0 \end{bmatrix}$$

and $\phi(z) = z^3$ is the nonlinearity applied element-wise. This function is not slope-restricted, so the simple attractor does not satisfy the assumptions of the Lurie network. The model is 3-contracting and has several attractor states.

**D.2 Training**

The default training settings common to the opinion, Hopfield and attractor datasets are presented in Table 5. The only parameters which varied between models were the learning rate and epoch which it was cut. Deviations from the default settings are detailed in Table 6. These values were chosen based on observations during training. No hyperparameter sweep was performed. The same details are also presented for the graph-coupled Hopfield network and graph-coupled attractor datasets in Table 7 and Table 8. When training the models for the opinion, Hopfield and attractor datasets, a single T4 GPU (accessed through Google Colab) was used. A single A100 GPU (also accessed through Google Colab) was used for training the models on the graph-coupled Hopfield and graph-coupled attractor datasets.

Table 5: Default training settings for opinion, Hopfield and attractor datasets.

| Parameter | Value |
|---|---|
| Batches | 10 |
| Batch size | 100 |
| Test split | 0.1 |
| Epochs | 100 |
| Criterion | Mean squared error |
| Optimiser | Adam (default settings) |
| Learning rate (LR) | $1 \times 10^{-2}$ (no decay) |

Table 6: Deviations from the default settings for the opinion, Hopfield and attractor datasets. LR decay includes the size of the decay followed by the epoch the decay was made.

| Dataset | Model | LR | LR Decay |
|---|---|---|---|
| Opinion Dynamics | Lurie Network | $5 \times 10^{-3}$ | |
| Opinion Dynamics | Antisymmetric RNN | $5 \times 10^{-3}$ | - |
| Hopfield Network | $k$-Lurie Network | $5 \times 10^{-3}$ | - |
| Hopfield Network | Neural ODE | $1 \times 10^{-3}$ | $0.1, 75$ |
| Hopfield Network | Antisymmetric RNN | $5 \times 10^{-3}$ | - |
| Simple Attractor | Neural ODE | $1 \times 10^{-3}$ | - |
| Simple Attractor | Antisymmetric RNN | $5 \times 10^{-3}$ | - |

Table 7: Default training settings for the graph-coupled Hopfield and attractor datasets.

| Parameter | Value |
|---|---|
| Batches | 15 |
| Batch size | 2000 |
| Test split | $\frac{1}{15}$ |
| Epochs | 100 |
| Criterion | Mean squared error |
| Optimiser | Adam (default settings) |
| Learning rate (LR) | $5 \times 10^{-3}$ (no cuts) |

Table 8: Deviations from the default settings for the graph-coupled Hopfield and attractor datasets. LR decay includes the size of the decay followed by the epoch the decay was made.

| Dataset | Model | LR | LR decay |
|---|---|---|---|
| GC Hopfield Network | Lipschitz RNN | $5 \times 10^{-3}$ | 0.2, 60 |
| GC Hopfield Network | GLN | $3 \times 10^{-3}$ | $\frac{1}{3}$, 60 |
| GC Simple Attractor | GLN | $1 \times 10^{-2}$ | - |
| GC Simple Attractor | $k$-Lurie Network | $1 \times 10^{-2}$ | - |
| GC Simple Attractor | Neural ODE | $5 \times 10^{-3}$ | 0.5, 40 |

### D.3 Further Results

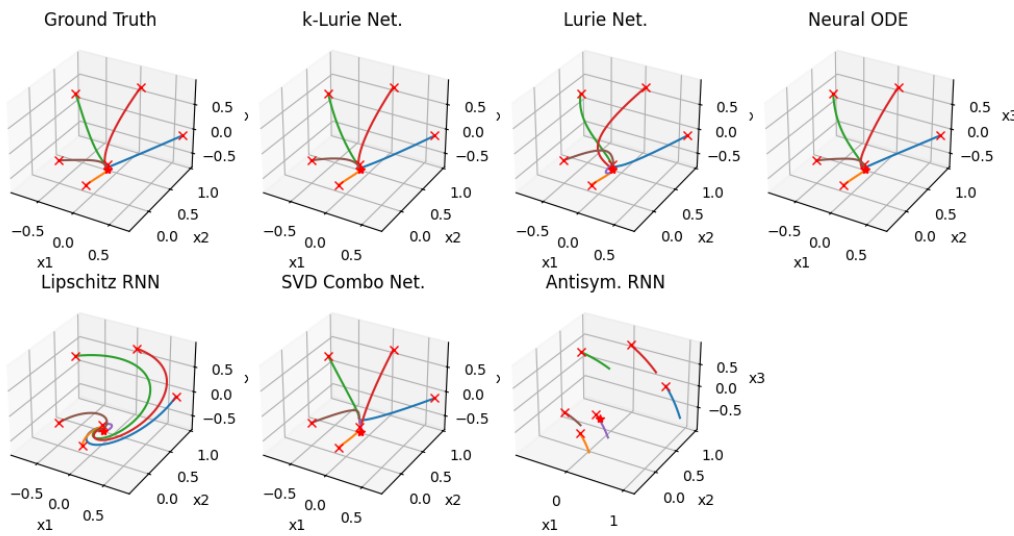

Figure 3: Random sample of trajectories from the opinion dynamics test set. Predictions are made by the model associated with the best MSE in Table 1. Crosses denote initial conditions, stars denote equilibrium points.

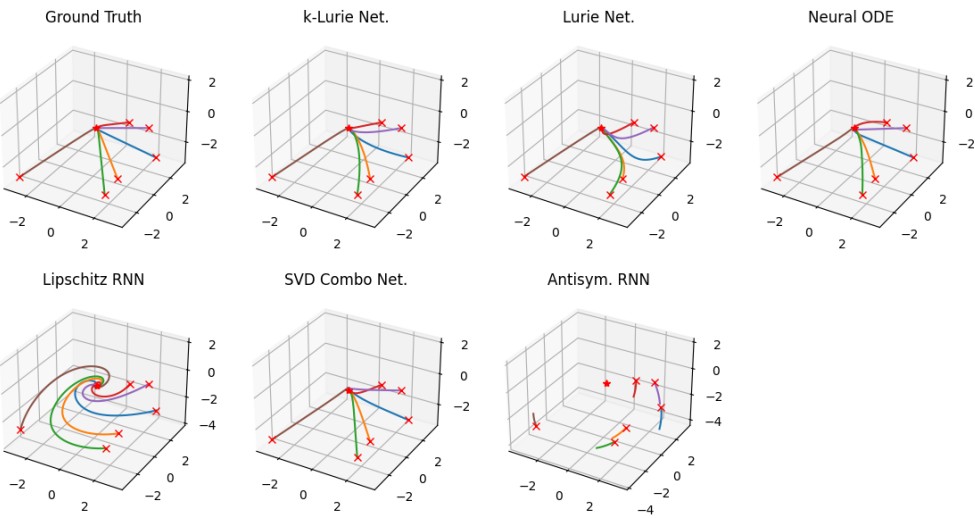

Figure 4: Random sample of 30$s$ trajectories from the noise-free, out of distribution opinion dynamics dataset. Predictions are made by the model associated with the best MSE in Table 1. Crosses denote initial conditions, stars denote equilibrium points.

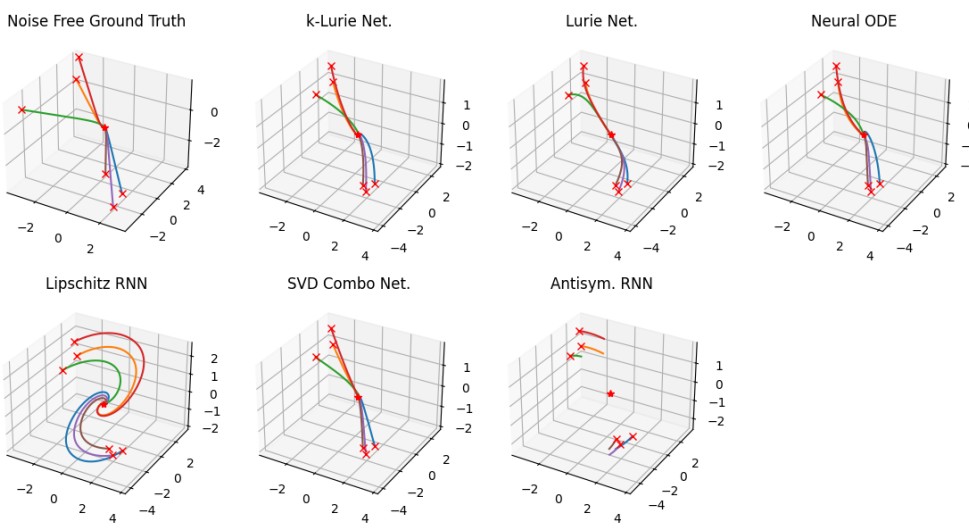

Figure 5: Random sample of 30$s$ trajectories from the noisy, out of distribution opinion dynamics dataset. Predictions are made by the model associated with the best MSE in Table 1. Crosses denote initial conditions, stars denote equilibrium points.

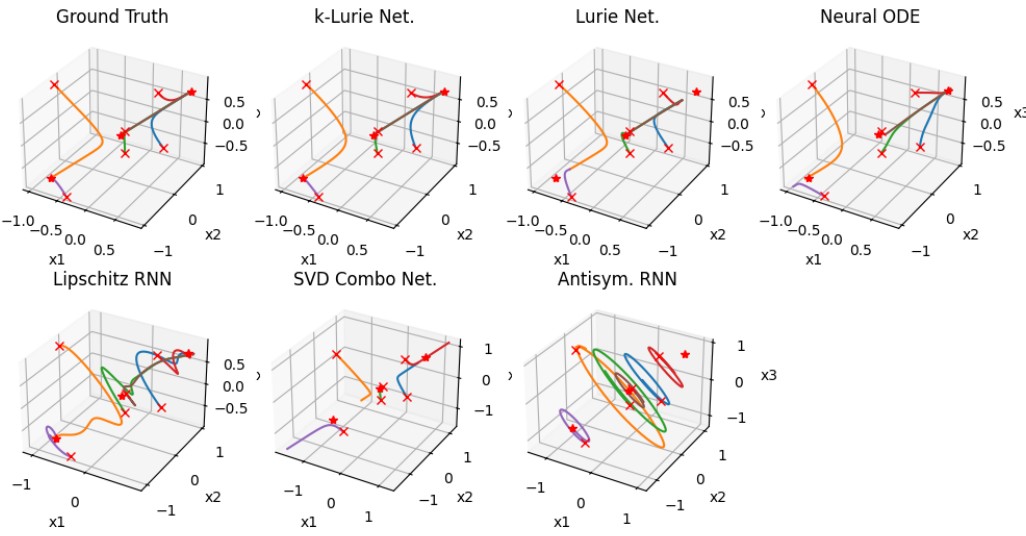

Figure 6: Random sample of trajectories from the Hopfield network test set. Predictions are made by the model associated with the best MSE in Table 1. Crosses denote initial conditions, stars denote equilibrium points.

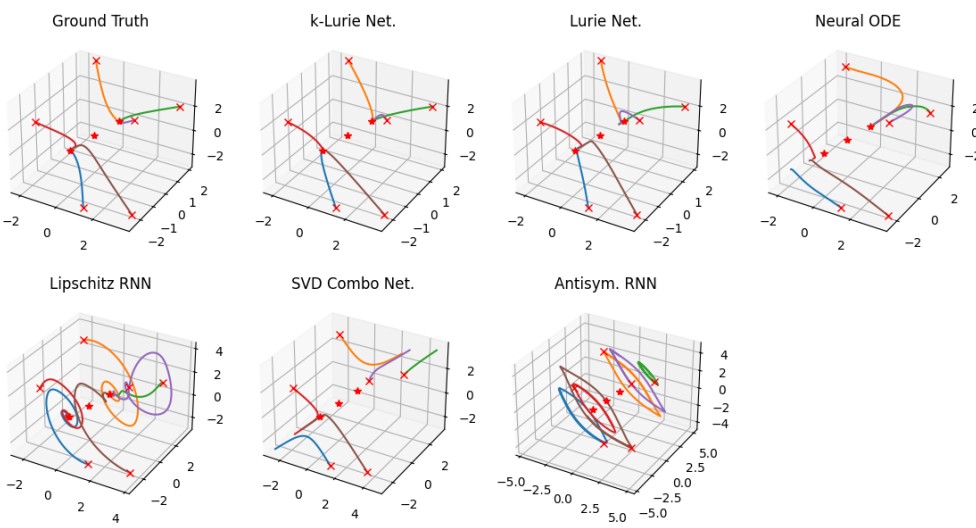

Figure 7: Random sample of $30s$ trajectories from the noise-free, out of distribution Hopfield network dataset. Predictions are made by the model associated with the best MSE in Table 1. Crosses denote initial conditions, stars denote equilibrium points.

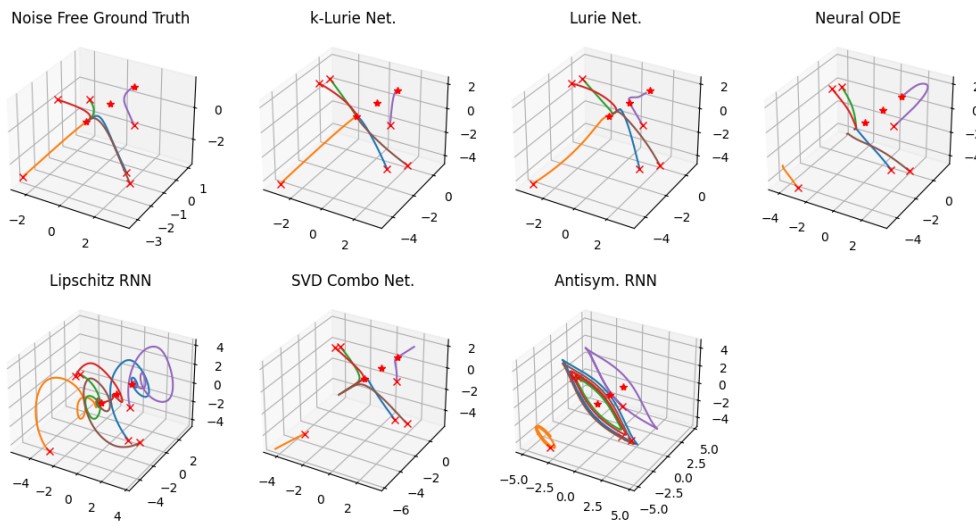

Figure 8: Random sample of $30s$ trajectories from the noisy, out of distribution Hopfield network dataset. Predictions are made by the model associated with the best MSE in Table 1. Crosses denote initial conditions, stars denote equilibrium points.

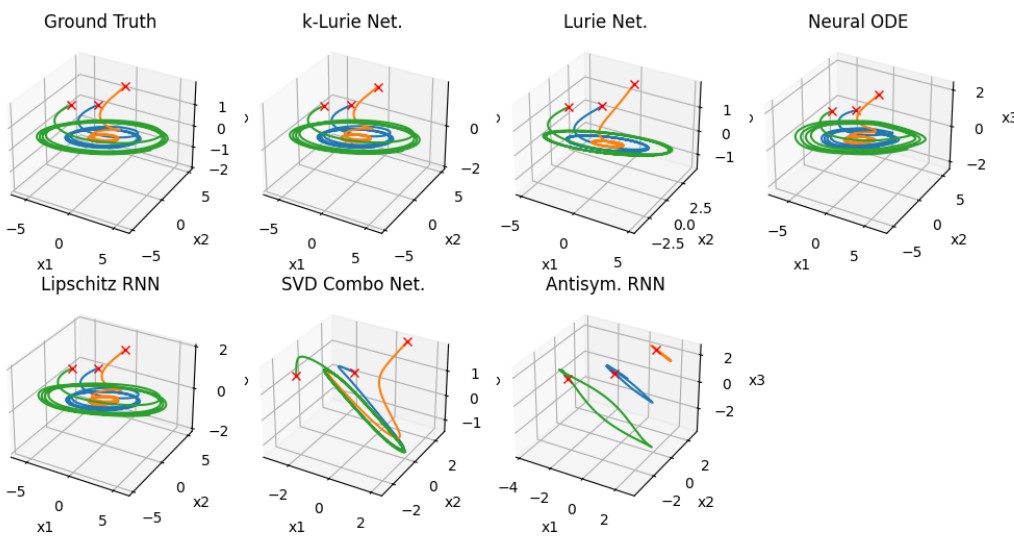

Figure 9: Random sample of trajectories from the simple attractor test set. Predictions are made by the model associated with the best MSE in Table 1. Crosses denote initial conditions.

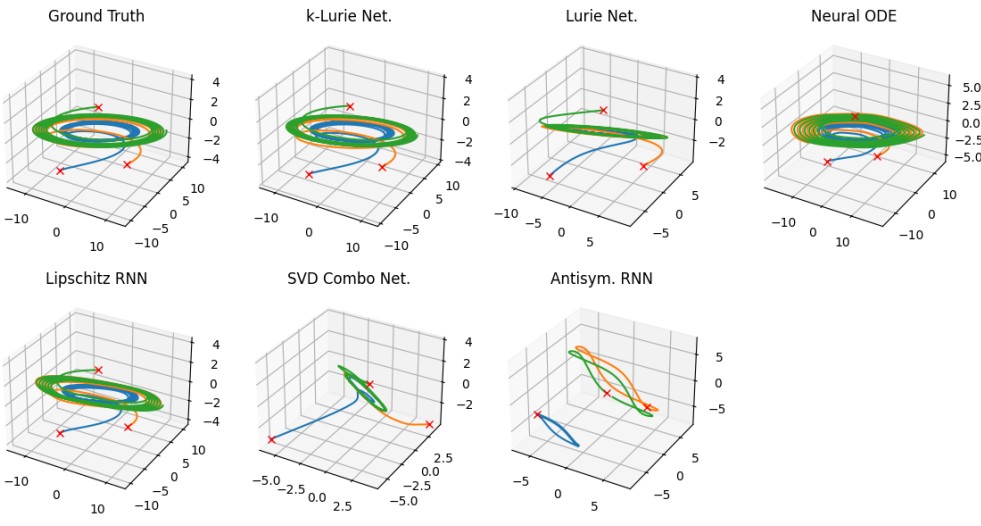

Figure 10: Random sample of 30$s$ trajectories from the noise-free, out of distribution simple attractor dataset. Predictions are made by the model associated with the best MSE in Table 1. Crosses denote initial conditions.

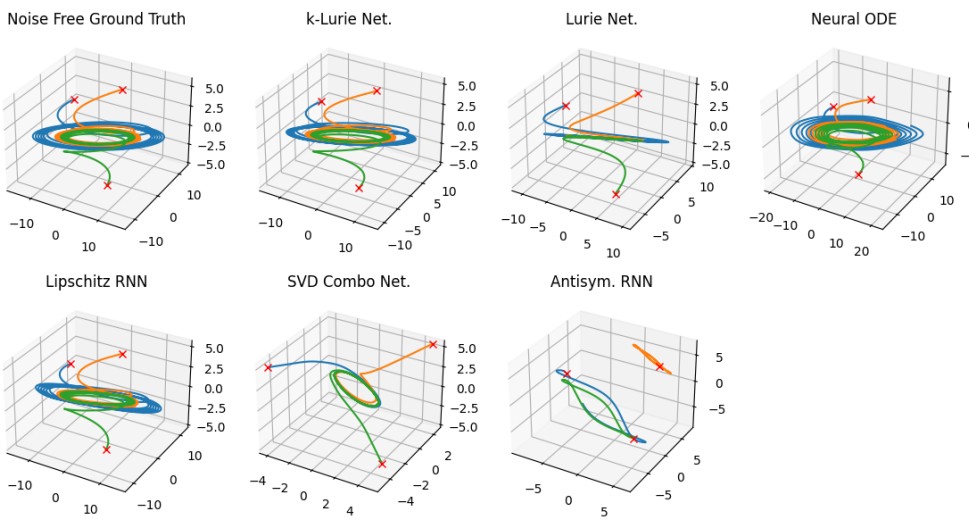

Figure 11: Random sample of 30$s$ trajectories from the noisy, out of distribution simple attractor dataset. Predictions are made by the model associated with the best MSE in Table 1. Crosses denote initial conditions.

