# OpenReview forum: "Lurie Networks with Robust Convergent Dynamics"
_TMLR — Accepted by TMLR_

### Review · Reviewer_8uBL · 2025-02-10

**Summary Of Contributions:**

Authors propose a Neural ODEs called Lurie Network. A parametric form of Lurie Networks encompasses several classes of continuous RNNs, Hopfield networks, neural oscillators, etc. To ensure stability authors use k-contraction analysis and derive several convenient parametrisations that guarantee k-contraction property for Lurie Network. It is demonstrated on the set of synthetic benchmarks that constrained Lurie Networks are competitive with other architectures.

**Audience:**

Yes

**Claims And Evidence:**

Yes

**Requested Changes:**

1. **Clarity and organisation**
   1. Authors gather theoretical facts and definitions in the appendix. Unfortunately, the main text of the article does not reference appendix too often. Below are several examples where a reference to the appendix would be appropriate:
      1. [Section 2.1] $k-$multiplicative and $k$-additive compound of matrix are not widely known, consider adding references to Definition 1 and Definition 2 in Appendix A
      2. [Theorem 1] It is hard to understand the content of Theorem 1 without explicit definition of what $k$-contraction means.
   2. Examples of Lurie networks in Section 3.1. are all RNNs. Can the authors please provide some other examples, e.g., Hopfield networks and Linear State Space Models (mentioned in the introduction), Neural Oscillators (mentioned in the abstract), etc?

2. **k-contraction is too restrictive and not selective enough**
   1. It seems to me that $k$-contraction is convenient but too restrictive. I think this should be discussed in the article. I will provide several examples:
      1. $2$-contraction illustrated in Figure 2 implies that all steady states of Hopfield network with this property are confined to the line. It seems that this condition is too restrictive for associative memory to be practically useful. Given that it is unfair to claim in the Section 6 that "A 2-contracting model can replicate the behaviour of associative memory, where every stored pattern corresponds to an equilibrium".
      2. If one applies Theorem 2 to the Lurie network with $A=-I$, the condition for $1$-contraction is $\\left\\|B\right\\|_{F} < \frac{1}{L}$ where $L$ is Lipschitz constant of nonlinearity used. It seems that under a much milder sufficient condition $\\left\\|B\\right\\|_2<\frac{1}{L}$ the resulting ODE still has a single global steady state.
   2. I also think that $k$-contraction is not selective enough. I am not entirely sure, but from Theorem 2 it looks like $k+1$-contraction implies $k$-contraction. If this is the case, $3$-contraction from Figure 1 can also be used to learn Hopfield network (with memories restricted to the line) and contraction with single globally stable steady state.

3. **Numerical part**
   1. I think that training time provided in Table 1 and Table 2 is misleading. I performed my own experiments with Hopfield data provided by authors. The results can be found in https://colab.research.google.com/drive/1arYUr0RkWGR-NH8mT_o-aUHtzdQPmUfO?usp=sharing. What I observed for NeuralODE:
      1. For batch size $100$, $100$ epoch will take about $12$ minutes on GPU and $8$ minutes on CPU using jax, diffrax, optax and equinox.
      2. Training time will depend on the batch size, solver, number of checkpoints, adjoint method, optimiser, precise details of architecture, hardware used, etc. It is virtually impossible to claim that one method is superior to the other one: it will require very expensive gridsearch on the space of hyperparameters.

	 Beside these details, there are also different training strategies that are more suitable for NeuralODEs. For example, one can train with flow matching avoiding direct solution of ODE on the training phase. In my view, all that makes it hard to perform a fair comparison of different methods, so I suggest extending discussion on this and removing training time from the tables.

   2. What is the reason Neural ODE has a much larger number of parameters? Why are there no experiments on the influence of network size? This information is of interest also for the out of distribution generalisation discussed in Table 2.

   4. There is an implicit assumption that we know which Lurie network to use for a particular dataset. What happens if we use a $1$-contracting or $3$-contracting network for Hopfield dataset? What happens if we use a $2$-contracting network for an attractor dataset? What happens if Hopfield dataset has three attractors not confined to a single line but we still use the $2$-contracting Lurie network?

   5. One claim of the authors is that constraints of $k$-contractive Lurie networks are beneficial for accuracy, robustness, and numerical efficiency. However, from the empirical data it is not clear whether the constraints help in learning and that they are beneficial for out-of-distribution generalisation. If one compares Table 1 and Table 2, contractive Lurie network is two orders less accurate on Option and Attractor datasets and $5\times$ less accurate for Hopfield dataset. Unconstrained Lurie network lost one order of accuracy for both Option and Attractor and performed better on Hopfield dataset. Given that some may argue that unconstrained Lurie network is more robust. It is also not clear why the unconstrained Lurie network has fewer parameters than the $k$-contractive one. The same goes for all other benchmarks: unconstrained Lurie network is smaller and clearly undertrained.

   7. Another interesting part in comparison of unconstrained Lurie and constrained Lurie networks is that they are implemented differently. In the code authors provided they use all kinds of matrices with special structures to define parameters of $k$-contractive Lurie network and just linear layers to implement unconstrained Lurie networks. The difference in the parametrisation leads to distinct training dynamics. I suggest authors use the same parametrisation (and the same number of parameters) for both constrained and unconstrained Lurie networks to remove the difference between training dynamics. For example since in constrained Lurie network authors represent matrix $B$ as $U_B\Sigma_B V_B^\top$, the same expression should be used to parametrise $B$ in the unconstrained Lurie networks since SVD is unique (up to a complex sign) and exist for arbitrary matrix.

**Strengths And Weaknesses:**

Strengths:
1. Authors introduce and explain techniques from the control theory that are not well-known in the ML community. I believe these techniques can be beneficial for other researchers.
2. The motivation is well-articulated and the organisation of the paper is clear enough to understand the essential ideas.
3. Obtained theoretical results are well-suited for practical realisation.
4. Provided code and comments in the appendix allows for reproduction of results.
5. Appendix contains many technical details and definitions that can be useful for readers unfamiliar with k-contraction analysis.

Weaknesses (explained in details in the section below):
1. k-contraction property seems too restrictive especially for Hopfield networks and for some cases it is not selective enough
2. Numerical part of the article is not convincing
3. There are some problems with organisation of the text

---

### Review · Reviewer_AryQ · 2025-02-19

**Summary Of Contributions:**

This paper presents an thorough analysis of the k-contraction properties of Lurie Networks, an important class of nonlinear models which appear across many different domains. The paper shows that Lurie Networks which are parameterized to be k-contracting have favorable (and sometimes superior) performance properties compared to 1-contracting networks trained on the same task.

**Audience:**

Yes

**Broader Impact Concerns:**

None.

**Claims And Evidence:**

Yes

**Requested Changes:**

None.

**Strengths And Weaknesses:**

The paper is very well-written, with clear motivation. The mathematical proofs were easy to follow, and the final theorems are intuitive. I do not see any weaknesses, given the target audience of this paper.

---

### Review · Reviewer_cQ12 · 2025-03-05

**Summary Of Contributions:**

This paper introduces the Lurie Network as a unifying architecture for modelling time-invariant nonlinear dynamical systems.  Several typical architectures, such as recurrent neural networks (RNNs) and neural oscillators, can be described using the Lurie Networks as special cases.
This paper rigorously derives sufficient conditions under which a Lurier Network has k-contraction properties.  This result provides a theoretical guarantee of global convergence to a point, line, or plane in the state-space, and extends the applicability of stability analysis to broader classes of dynamical systems.
This paper also derives a special parametrization of the Lurie  Network, ensuring the Lurie Network possesses the k-contraction property. Also this parametrization is unconstrained and enables simple gradient-based training of the Lurier Network with k-contraction property.

This paper also propose the Graph Lurier Network (GLN) to capture interactions between interconnected subsystems possesing the k-contraction property.  This paper provides the corresponding theoretical results for GLN.

This paper presents empirical results that imply the effectiveness of the Lurie Network with the special parametrization and k-contraction property.  This paper provides the numerical evidence of the high-performance in learning problem of dynamical systems in comparison with other existing architectures, for example, Neural ODE.

**Audience:**

Yes

**Claims And Evidence:**

Yes

**Requested Changes:**

- Since chaotic systems are inherently unpredictable over long horizons,  the long-term prediction conducted in this paper will not work, however, qualitative discussions (e.g., how k-contraction might capture local stability properties or the limits of predictability) would strengthen the contribution.

- Although real-data experiments might be beyond the scope of this paper, a discussion on how the Lurie Network and GLN could be applied to real data in neuroscience would improve the paper.

### Minor error
- In Section 2.1, the authors use $\mathcal{SO}(n)$ and $so(n)$ to describe the orthogonal and skew-symmetric matrices, but they are usually denote the special orthogonal group (orthogonal matrices with determinant 1) and its Lie algebra.  The author should follow typical notation of the theory of Lie group.
- p.5, l.5, there is double "it".
- p.18, in the formulas in center, use roman $\exp$ not italic $exp$.

**Strengths And Weaknesses:**

### Strengths
- Introduction the Lurie Network as a unifying architecture for modeling nonlinear dynamical systems, covering existing models such as RNNs and Neural Oscillators.
- Introduction of k-contraction as a general stability measure, enabling global convergence to a point, line, or plane in the neural state space.
- Rigorous sufficient condition for Lurie Network and Graph Lurier Network (GLN) to be k-contracting.
- Introduction of unconstrained parametrizations that allow the k-contracting Lurie Network and GLN to be trained with standard gradient-based optimization algorithms.

### Weaknesses
- Lack of discussion on chaotic systems: the paper does not explicitly discuss chaotic systems, which are crucial in many nonlinear dynamical applications, for example, the Lorenz attractor is 3-contracting, but there is no discussion on whether the proposed framework can model or analyze such systems.
- Experimental validation limited to toy datasets: While neuroscientific applications and real-world systems are mentioned in Introduction, the experiments are conducted only on toy datasets.

---

### Author Response · Authors · 2025-03-18
**Response to Reviewer cQ12**

We would like to thank Reviewer cQ12 for the kind and helpful review. We have addressed your individual comments below.

> Since chaotic systems are inherently unpredictable over long horizons, the long-term prediction conducted in this paper will not work, however, qualitative discussions (e.g., how k-contraction might capture local stability properties or the limits of predictability) would strengthen the contribution.

We have added the following discussion at the end of the first paragraph on page 11: This includes certain types of chaotic systems, for example Thomas' cyclically symmetric attractor. At the edge of the chaotic regime, the trajectories converge to a strange attractor, which could be modelled by a $3$-contracting Lurie Network.

> Although real-data experiments might be beyond the scope of this paper, a discussion on how the Lurie Network and GLN could be applied to real data in neuroscience would improve the paper.

We agree that this is an important point, but beyond the scope of this work. However, we did include some discussion of how we think this could be an interesting framework for modelling memory retrieval tasks in the fourth paragraph of the related work. Is there something more specific you were thinking of?

> Minor errors

We have corrected all minor errors mentioned. The notation $\mathcal{O}(n)$ and ${\rm Skew}(n)$ are now used to respectively denote the sets of orthogonal and skew-symmetric matrices.

---

### Author Response · Authors · 2025-03-18
**Response to Reviewer AryQ**

We would like to thank Reviewer AryQ for the kind and generous review. We are thrilled to hear you were so happy with the paper.

---

### Author Response · Authors · 2025-03-18
**Response 1 to Reviewer 8uBL**

Thank you to Reviewer 8uBL for the detailed and constructive feedback. We have responded to your comments below.

> Authors gather theoretical facts and definitions in the appendix. Unfortunately, the main text of the article does not reference appendix too often. Below are several examples where a reference to the appendix would be appropriate:
> - Section 2.1: multiplicative and additive compound of matrix are not widely known, consider adding references to Definition 1 and Definition 2 in Appendix A.
> - Theorem 1: It is hard to understand the content of Theorem 1 without explicit definition of what $k$-contraction means.

Thank you for bringing this to our attention, we have added several references to the Appendix throughout Section 2.

> Examples of Lurie networks in Section 3.1. are all RNNs. Can the authors please provide some other examples, e.g., Hopfield networks and Linear State Space Models (mentioned in the introduction), Neural Oscillators (mentioned in the abstract), etc?

To avoid disrupting the flow of the paper, we included these examples in Appendix B.2. We opted to include the RNN examples in the main text as we compared against these in the empirical evaluation.

> $2$-contraction illustrated in Figure 2 implies that all steady states of Hopfield network with this property are confined to the line. It seems that this condition is too restrictive for associative memory to be practically useful. Given that it is unfair to claim in the Section 6 that ``A 2-contracting model can replicate the behaviour of associative memory, where every stored pattern corresponds to an equilibrium".

We agree that this is a limitation for this application and have added some additional comments in Section 6. However, we don't think this necessarily means the condition is too restrictive to be practically useful given the line is infinite. Furthermore, by using our results, the Lurie network does not require the parameters to be symmetric. This biologically unrealistic simplification is typically needed for Hopfield-based models. We have also added a comment in Section 3.2 to highlight this.

> If one applies Theorem 2 to the Lurie network with $A = -I$, the condition for $1$-contraction is $|| B ||_{F} < \frac{1}{L}$ where $L$ is the Lipschitz constant of nonlinearity used. It seems that under a much milder sufficient condition the resulting ODE still has a single global steady state.

We agree that less conservative results can be obtained for the special case of $1$-contraction. However, the main novelty of our approach is that it can be used to verify and encode three types of convergent dynamics. Those including multiple equilibrium points and limit cycles cannot be so easily studied by other means and are more interesting for ML applications. Furthermore, for the example you have given (if we understand correctly), we believe the condition for $1$-contraction when applying Theorem 2 should be  $||B||_2 < \frac{1}{L}$ which is less conservative than the condition above since $||B||_2  < ||B||_F$.

> I also think that $k$-contraction is not selective enough. I am not entirely sure, but from Theorem 2 it looks like $(k+1)$-contraction implies $k$-contraction. If this is the case, $3$-contraction from Figure 1 can also be used to learn Hopfield network (with memories restricted to the line) and contraction with single globally stable steady state.

Theorem 2 is used to verify if a given Lurie network is $k$-contracting. In fact if Theorem 2 verifies a system is $k$-contracting, by definition, that same system must also be $l$-contracting, where $l \geq k$. In Theorem 5 and Theorem 6, the user can select $k$ to specify which type of $k$-contracting Lurie networks can be optimised over during training. For example, if the user set $k=3$, the Lurie network would be optimised over $1$, $2$ and $3$ contracting Lurie networks. If the user set $k=2$, the Lurie network would be optimised over $1$ and $2$ contracting Lurie networks. These results become more selective as the value of $k$ is reduced. This is a slight limitation if the user knows the value of $k$ in advance; however, one would expect the model to quickly learn from the data which type of $k$-contracting system is most appropriate since the trajectory data of a $2$-contracting system would be very different to that of a $1$ or $3$ contracting system. Finally, our approach has significant upside if the value of $k$ is unknown. In this case, the user could set $k=3$ and search over many $(1,2,3)$-contracting systems. We have added a comment to Section 4 (just above Theorem 7) to highlight this. We also added a proof of the intersection condition to the appendix.

---

### Author Response · Authors · 2025-03-18
**Response 2 to Reviewer 8uBL**

> I think that training time provided in Table 1 and Table 2 is misleading. I performed my own experiments with Hopfield data provided by authors. What I observed for NeuralODE ...

Thank you for raising this point, we have removed the training times from the table.

> What is the reason Neural ODE has a much larger number of parameters?

The Neural ODE was the only purely black-box model, hence we allowed it to have a larger number of parameters. Each of the other models was a special case of the Lurie network with different stability constraints.

> There is an implicit assumption that we know which Lurie network to use for a particular dataset. What happens if we use a $1$-contracting or $3$-contracting network for Hopfield dataset? What happens if we use a $2$-contracting network for an attractor dataset? What happens if Hopfield dataset has three attractors not confined to a single line but we still use the $2$-contracting Lurie network?

If the value of $k$ set by the user was below the true value of $k$, this would prevent the network from learning an accurate model of the data. However, if the value of $k$ is unknown, this can be handled by setting $k=3$ as this allows the model to train over $1$, $2$, $3$-contracting systems.

> Another interesting part in comparison of unconstrained Lurie and constrained Lurie networks is that they are implemented differently. In the code authors provided they use all kinds of matrices with special structures to define parameters of $k$-contractive Lurie network and just linear layers to implement unconstrained Lurie networks. The difference in the parametrisation leads to distinct training dynamics. I suggest authors use the same parametrisation (and the same number of parameters) for both constrained and unconstrained Lurie networks to remove the difference between training dynamics. For example since in constrained Lurie network authors represent matrix $B$ as $U_{B} \Sigma_{B} V_{B}^{\top}$, the same expression should be used to parametrise $B$ in the unconstrained Lurie networks since SVD is unique (up to a complex sign) and exist for arbitrary matrix.

From a representational perspective, this should not make any difference as both representations should be equivalent in the unconstrained case. However, we had not considered the impact this would have on training dynamics. We implemented the unconstrained Lurie network as requested and observed the following results.

      |   Model     |           MSE (mean $\pm$ std, best)
      Lurie Network | $(1.7 \pm 2.5, 0.92) x 10^{-2}$ (Opinion)
      Lurie Network | $(3.4 \pm 2.3, 1.64) x 10^{-1}$ (Hopfield)
      Lurie Network | $1.30 \pm 2.0, 7.07 x 10^{-2}$ (Attractor)

      |   Model     |           MSE (mean $\pm$ std, best)
      Lurie Network | $4.9 x 10^{-1}$ (Opinion, OOD)
      Lurie Network | $7.13$ (Hopfield, OOD)
      Lurie Network | $37.32$ (Attractor, OOD)
      Lurie Network | $7.1 x 10^{-1}$ (Opinion, OOD + noise)
      Lurie Network | $9.61$ (Hopfield, OOD + noise)
      Lurie Network | $48.27$ (Attractor, OOD + noise)

Clearly, this change seems to negatively impact the final results which suggests unnecessarily parametrising the model makes the model more challenging to train. For this reason, we think it makes sense to use the original implementation of the unconstrained Lurie network and keep the original results.

> One claim of the authors is that constraints of $k$-contractive Lurie networks are beneficial for accuracy, robustness, and numerical efficiency. However, from the empirical data it is not clear whether the constraints help in learning and that they are beneficial for out-of-distribution generalisation. If one compares Table 1 and Table 2, contractive Lurie network is two orders less accurate on Opinion and Attractor datasets and $5 \times$ less accurate for Hopfield dataset. Unconstrained Lurie network lost one order of accuracy for both Opinion and Attractor and performed better on Hopfield dataset. Given that some may argue that unconstrained Lurie network is more robust.

 This is an interesting point and raises the question of whether absolute mean squared error or relative mean squared error is a better measurement of robustness. However, we believe our use of the absolute mean squared error is justified given that it is in line with popular robustness benchmarks, such as RobustBench. This benchmark ranks its models based on classification accuracy when under adversarial attack and does not consider the relative drop w.r.t its nominal classification accuracy.

---

> ### Comment · Reviewer_8uBL · 2025-04-04
>
> I would like to thank the authors for a detailed reply and especially for conducting additional experiments. The rebuttal by authors sufficiently addresses my main concerns, so I believe the article is suitable for publication in TMLR.

---

### Author Response · Authors · 2025-03-18
**General Response**

Firstly, we would like to thank all the reviewers for their kind and helpful comments. Thanks to them, we have considered new implications of our research and believe we now have a more complete paper. In response to your comments, we have updated the manuscript. Some of the changes have been left as red text to help with reviewing.

---

### Decision · Action_Editor_Z3eH · 2025-04-16

**Recommendation:** Accept as is

**Comment:**

I recommend acceptance without revision. The paper introduces an elegant and general framework for learning stable neural dynamics via Lurie Networks, characterized by k-contracting properties. The authors provide strong theoretical underpinnings, including new sufficient conditions and an unconstrained parametrization that facilitates efficient training. Despite concerns about limitations in empirical breadth and application to real-world data, the response to reviewers was detailed and constructive, and the revised manuscript adequately addresses key points. The clear exposition and reproducibility further enhance its suitability for publication. While the practical significance may benefit from future real-world applications, the methodological contributions are substantial and timely.

**Audience:**

This work will be of strong interest to TMLR’s readership, particularly researchers in dynamical systems, neural ODEs, and stability-constrained learning. The introduction of k-contraction as a general stability criterion and the Lurie Network framework offers new analytical tools and design principles for building robust neural dynamical models. Its potential applications extend to memory modeling, neuroscience, and control, making the paper valuable not only to theoretical audiences but also to practitioners interested in stable and interpretable continuous-time models.

**Claims And Evidence:**

The submission presents a novel and rigorous formulation of Lurie Networks as a unifying architecture for modeling nonlinear dynamical systems through the lens of k-contraction. The theoretical claims are well-supported: sufficient conditions for k-contracting behavior are derived, and the resulting parametrization enables stable training via standard gradient-based methods. The authors further propose a graph-based extension and demonstrate both theoretical soundness and empirical effectiveness through comprehensive experiments. While the numerical evaluations focus on synthetic datasets, they are adequate for validating the core claims regarding stability, robustness, and representational capacity. The revisions in response to reviewer concerns, including clarification of constraints, additional discussion on chaotic systems, and architectural implementation details, significantly strengthen the evidence supporting the paper’s contributions.

---

> ### Author Response · Authors · 2025-04-19
> **Submitted Camera-Ready Version**
>
> Dear AE and Reviewers,
>
> We thank you for the insightful reviews and for accepting our paper.
>
> We have submitted the camera-ready version of our manuscript.